# Inside-Out: Hidden Factual Knowledge in LLMs

**Zorik Gekhman**[T]   **Eyal Ben David**[G]   **Hadas Orgad**[T]   **Eran Ofek**[G]

**Yonatan Belinkov**[T]   **Idan Szpector**[G]   **Jonathan Herzig**[G]   **Roi Reichart**[T]

[T]Technion - Israel Institute of Technology
[G]Google Research

zorikgekhman@gmail.com

## Abstract

This work presents a framework for assessing whether large language models (LLMs) encode more factual knowledge in their parameters than what they express in their outputs. While a few studies hint at this possibility, none have clearly defined or demonstrated this phenomenon. We first propose a formal definition of *knowledge*, quantifying it for a given question as the fraction of correct-incorrect answer pairs where the correct one is ranked higher. This gives rise to *external* and *internal* knowledge, depending on the information used to score individual answer candidates: either the model's observable token-level probabilities or its intermediate computations. Hidden knowledge arises when internal knowledge exceeds external knowledge. We then present a case study, applying this framework to three popular open-weight LLMs in a closed-book QA setup. Our results indicate that: (1) LLMs consistently encode more factual knowledge internally than what they express externally, with an average relative gap of 40%. (2) Surprisingly, some knowledge is so deeply hidden that a model can internally know an answer *perfectly*, yet fail to generate it even once, despite large-scale repeated sampling of 1,000 answers. This reveals fundamental limitations in the generation capabilities of LLMs, which (3) put a practical constraint on scaling test-time compute via repeated answer sampling in closed-book QA: significant performance improvements remain *inaccessible* because some answers are practically never sampled, yet if they were, we would be guaranteed to rank them first.[1]

## 1 Introduction

Large language models (LLMs) excel at knowledge-intensive tasks, yet fundamental questions remain open about the nature of their factual knowledge. What does it mean for LLMs to know a fact? And equally important, do LLMs store more factual information in their parameters than they express in their outputs? Answering the latter question rigorously can be highly valuable: if such knowledge is stored but not used, we might be able to develop methods to surface it, improving performance and reliability. From a safety perspective, undisclosed knowledge poses risks if it unexpectedly surfaces, revealing sensitive information or producing outputs never meant to be shared. Lastly, beyond practical considerations, this question is key to advancing interpretability: if models encode more knowledge than they express in their outputs, it highlights the need to understand how this knowledge is accessed or suppressed during inference.

In this work, our goal is to study whether LLMs encode more factual knowledge in their parameters than they express in their outputs, a phenomenon we refer to as *hidden knowledge*. So far, prior work has only hinted at its existence (Burns et al., 2023; Orgad et al., 2025), but none have clearly defined it or the conditions required to demonstrate its existence. Consequently, our core contribution is proposing a concrete definition that provides a framework for systematically studying hidden knowledge.

---

[1]https://github.com/zorikg/inside-out

To define hidden knowledge we first need to define *knowledge*, which is challenging since there is no well-defined notion of it for LLMs in the literature (Fierro et al., 2024). We couple knowledge with the ability to rank correct answers above incorrect ones using a scoring method (e.g., token-level likelihood) and quantify it per-question as the fraction of (correct, incorrect) answer pairs ranked correctly (§2.1). While other possible definitions could be proposed, ours has three key advantages. First, unlike existing perspectives on LLM's knowledge (Fierro et al., 2024), it is specified with a clear computational procedure, making it useful for empirical studies. Second, it addresses limitations in the common practice of measuring knowledge by evaluating performance of a single predicted answer on closed-book question-answering (QA) benchmarks (Petroni et al., 2019; Wei et al., 2024). Lastly, and most importantly, it is designed with the hidden knowledge research question in mind, ensuring that both the knowledge a model expresses *externally* and the knowledge it encodes *internally* are measured under a unified definition. Since we quantify knowledge relative to a function that assigns scores to candidate answers using information from the model, testing for hidden knowledge comes down to measuring knowledge through different scoring functions: *external* ones,[2] which are restricted to use only observable signals based on the model's token-level probabilities, and *internal* ones, which can use intermediate computations. Specifically, we define *hidden knowledge* as the existence of an internal function that ranks answers more accurately than any external function (§2.2).

Using this framework, we design a study using 1,700 closed-book QA questions. We estimate the set of (correct, incorrect) answer pairs per-question using 1,000 model-generated answers, labeled for correctness by an LLM judge that compares against the ground truth (Wei et al., 2024). For *internal* scoring, we train a linear classifier to predict correctness from the model's hidden representations of question-answer pairs (Hupkes et al., 2018; Belinkov & Glass, 2019). We then compare the *internal* knowledge measured by this classifier to *external* knowledge measured by popular scoring methods that use the model's token-level probabilities. Our results strongly indicate the presence of hidden knowledge. Across three LLMs, the internal scoring method consistently measures more knowledge than any external one, with all differences statistically significant and an average relative gap of 40%.

But how deeply can knowledge be hidden? In 56% of the questions, all the model-generated answers are wrong. So we manually add the ground-truth answer to the set of candidate answers, and analyze its impact on the model's internal knowledge scores. Surprisingly, in 9% of the questions, the internal scoring method scores the ground-truth answer higher than *any* incorrect candidate, despite the model *failing to generate it even once* across 1,000 attempts. This illustrates an extreme case of hidden knowledge: while models may occasionally generate an incorrect answer despite knowing the correct one (Simhi et al., 2024), it is highly surprising that a known correct answer is practically *never* generated, even with large scale repeated sampling. This highlights a fundamental limitation in the generation process of modern LLMs, which we hope future research on decoding mechanisms will explore.

Lastly, we leverage our findings to enhance closed-book QA performance. We obtain 12% average *relative* improvement over greedy decoding by increasing test-time compute through sampling a large set of answers and selecting the top one based on our internal scoring function. This extends existing evidence on the potential of using verifiers trained on LLMs hidden states to improve performance (Orgad et al., 2025). However, the more interesting insight is that there is potential for substantial gains of additional 40% relative improvement that remain *inaccessible* due to the constraints we identified in the LLM generation process: these constraints prevent some correct answers from even being sampled, yet if they were, we would be guaranteed to choose them since they would always be ranked first.

To conclude, we introduce a framework for evaluating the gap between the knowledge LLMs encode internally and what they express externally, and provide evidence of this gap in popular LLMs. Notably, its magnitude varies significantly: Gemma (Team et al., 2024) shows a relative gap of 57% but Llama (Dubey et al., 2024) 14%, highlighting a crucial direction for future work: understanding these differences and developing methods to surface internal knowledge more effectively, leading to more transparent and reliable LLMs.

---

[2]The term *external* may suggest elements outside the model, which is not our intention. An alternative name could be *observable*. We use *external* since it reads more smoothly in our context.

## 2 Hidden Knowledge

In this section, we tackle the challenge of evaluating hidden factual knowledge in LLMs, going beyond prior discussions (§5) by proposing a formal definition. We focus on knowledge represented as (subject, relation, object) triplets, whose structured nature simplifies evaluation and helps control for confounding factors such as ambiguous or easily guessable answers and overlaps between training and test examples. We first define *knowledge* (§2.1), focusing on subjects and relations with a *unique* object for clarity, as extension to multiple objects is straightforward. We then define the conditions under which a model is said to possess hidden factual knowledge (§2.2) and outline how to estimate it for a given LLM (§2.3).

### 2.1 Defining Knowledge Relative to an Answer Scoring Method

We consider an auto-regressive language model $M$, where the next-token distribution given a context $x$ is $P_M(\cdot \mid x)$. Given a question $q$ and an answer $a$, we denote $M$'s token-level probability of generating $a$ as $P_M(a \mid q) = \prod_{i=1}^{n} P_M(a_i \mid q, a_{<i})$. In practice, $M$ is prompted to answer $q$ (see §A.2), so $q$ denotes $\mathrm{Prompt}(q)$, but we omit the Prompt notation for simplicity.

A common way to test if $M$ knows a fact, e.g., *("Empire State Building", location, "NYC")*, is to prompt it with a question like *"Where is the Empire State Building located?"* (Wei et al., 2024). However, $M$'s outputs may vary with decoding strategy or question phrasing. Furthermore, decoding algorithms can lead to an incorrect answer despite $M$ assigning higher likelihood to a correct one. Lastly, one correct answer does not guarantee full knowledge, e.g., $M$ should know that both *"NYC"* and *"New York City"* are correct. This calls into question the reliability of inferring $M$'s knowledge from one answer. We propose to examine scores, such as token-level probabilities, that $M$ assigns to all plausible answer candidates. Specifically, we quantify knowledge per-question relative to the ability to score *any* correct answer higher than *any* plausible incorrect one, regardless of question phrasing. Since there are many possible ways to use $M$ to score an answer, we define knowledge *relative to a scoring method*, which is also useful for our objective of comparing between *internal* parametric knowledge, and the knowledge that is expressed *externally* (§2.2). We focus on a question-answer structure since it simplifies the definition and discuss alternative settings in §A.15.

**Definition 1** (Knowledge of a Model w.r.t a Scoring Method).
*For a model* $\mathbf{M}$ *and a fact represented as a* (subject, relation, object) *triplet* $(\mathbf{s}, \mathbf{r}, \mathbf{o})$, *e.g.,* ("France", capital, "Paris"), *we define the following sets:*

- $\mathbf{Q}(\mathbf{s}, \mathbf{r})$: *All paraphrases of questions based on* $\mathbf{s}$ *and* $\mathbf{r}$. *E.g., if* $(\mathbf{s}, \mathbf{r}) = $ ("France", capital), *it may include "What is the capital of France?", "Which city is the capital of France?", etc.*

- $\tilde{\mathbf{A}}(\mathbf{o})$: *All plausible answers to* $\mathbf{Q}(\mathbf{s}, \mathbf{r})$, *defined as all paraphrases of entities that have the same type as* $\mathbf{o}$. *E.g., if* $\mathbf{o} = $ *"Paris", it may include paraphrases of city names such as "Paris", "The city of New York", etc.*

- $\mathbf{A}(\mathbf{o}) \subseteq \tilde{\mathbf{A}}(\mathbf{o})$: *All paraphrases of* $\mathbf{o}$. *E.g., if* $\mathbf{o} = $ *"Paris", it may include "Paris", "The city of Paris", etc.*

- $\Omega(\mathbf{s}, \mathbf{r}, \mathbf{o}) := \mathbf{A}(\mathbf{o}) \times \left[ \tilde{\mathbf{A}}(\mathbf{o}) \setminus \mathbf{A}(\mathbf{o}) \right]$: *All ordered pairs of a correct answer and a plausible incorrect answer to* $\mathbf{Q}(\mathbf{s}, \mathbf{r})$. *E.g., if* $(\mathbf{s}, \mathbf{r}) = $ ("France", capital), *it may include ("Paris", "London"), ("Paris city", "NYC"), etc.*

*Given a scoring function* $\mathbf{S}_\mathbf{M} : \mathbf{Q}(\mathbf{s}, \mathbf{r}) \times \tilde{\mathbf{A}}(\mathbf{o}) \to \mathbb{R}$, *which, given a question answer pair* $(\mathbf{q}, \mathbf{a})$, *uses* $\mathbf{M}$ *to predict whether* $\mathbf{a}$ *is a correct answer to* $\mathbf{q}$,[3] *we define a per-question score* $\mathbf{K}_\mathbf{q}(\mathbf{s}, \mathbf{r}, \mathbf{o}; \mathbf{S}_\mathbf{M})$ *to quantify the ability to rank correct answers above plausible incorrect ones. Formally:*

$$\mathbf{K}_\mathbf{q}(\mathbf{s}, \mathbf{r}, \mathbf{o}; \mathbf{S}_\mathbf{M}) = \frac{1}{|\Omega(\mathbf{s}, \mathbf{r}, \mathbf{o})|} \sum_{(\mathbf{a}, \tilde{\mathbf{a}}) \in \Omega(\mathbf{s}, \mathbf{r}, \mathbf{o})} \mathbb{I}\big( \mathbf{S}_\mathbf{M}(\mathbf{q}, \mathbf{a}) > \mathbf{S}_\mathbf{M}(\mathbf{q}, \tilde{\mathbf{a}}) \big) \tag{1}$$

*The overall knowledge degree of* $\mathbf{M}$ *for the fact* $(\mathbf{s}, \mathbf{r}, \mathbf{o})$ *according to* $\mathbf{S}_\mathbf{M}$ *is then defined as:*

---

[3] $\mathbf{S}_M$ accounts for the different ways to score based on information from $M$. In case it has parameters (e.g., a probing classifier), we assume that they were not optimized using information about $(\mathbf{s}, \mathbf{r}, \mathbf{o})$.

$$\mathbf{K}(\mathbf{s}, \mathbf{r}, \mathbf{o}; \mathbf{S}_{\mathrm{M}}) = \frac{1}{|\mathbf{Q}(\mathbf{s}, \mathbf{r})|} \sum_{\mathbf{q} \in \mathbf{Q}(\mathbf{s}, \mathbf{r})} \mathbf{K_q}(\mathbf{s}, \mathbf{r}, \mathbf{o}; \mathbf{S}_{\mathrm{M}}) \tag{2}$$

*Finally, we define* $\mathbf{K}^*$ *to reflect cases of perfect knowledge, where* $\mathbf{S}_{\mathrm{M}}$ *correctly ranks all pairs:*

$$\mathbf{K}^*(\mathbf{s}, \mathbf{r}, \mathbf{o}; \mathbf{S}_{\mathrm{M}}) = \mathbb{I}\Big( \mathbf{K}(\mathbf{s}, \mathbf{r}, \mathbf{o}; \mathbf{S}_{\mathrm{M}}) = 1 \Big) \tag{3}$$

Since we quantify $\mathbf{K_q}$ based on answer from $\Omega(\mathbf{s}, \mathbf{r}, \mathbf{o})$, $\mathbf{S}_{\mathrm{M}}$ could in theory score an *implausible* answer (e.g., "#%") higher than a correct one. While it is extremely unlikely, we can also enforce it formally. We present an extended definition supporting this in §A.9. We also note that $\mathbf{K_q}$ is conceptually related to the area under the ROC curve (AUC-ROC), as one interpretation of AUC-ROC is the probability that a random positive answer is ranked higher than a random negative one (Fawcett, 2006). We discuss the differences in §A.11.

## 2.2 Evidence of Hidden Knowledge

We define hidden knowledge as cases where *M* embeds more knowledge in its parameters than it expresses externally. To formalize this, we differentiate between *internal* and *external* scoring functions. *External* functions are restricted to using only *M*'s observable signals based on its predicted token-level probabilities, while *internal* functions can leverage intermediate computations, such as hidden states. Hidden knowledge is measured by comparing the knowledge captured by internal vs. external scoring functions.

**Definition 2** (Evidence of Hidden Knowledge).
*Given a model* $\mathbf{M}$*, let* $\mathcal{T}_{\mathrm{M}}$ *be an internal scoring function,* $\mathcal{S}_M^E$ *the set of all external scoring functions, and* $\mathcal{D} = \big\{ (\mathbf{s}_i, \mathbf{r}_i, \mathbf{o}_i) \big\}_{i=1}^n$ *a dataset where each element is a unique fact. We say that* $\mathcal{T}_{\mathrm{M}}$ *demonstrates that* $\mathbf{M}$ *has* hidden knowledge *of* $\mathcal{D}$*, if* $\mathbf{M}$ *has more knowledge of* $\mathcal{D}$ *according to* $\mathcal{T}_{\mathrm{M}}$ *than according to any* $\mathcal{S}_M \in \mathcal{S}_M^E$*, with a margin* $\Delta$ *that ensures the difference is sufficiently large to rule out insignificant variations. Formally it is defined as follows:*

$$\frac{1}{n} \sum_{i=1}^n \mathbf{K}(\mathbf{s}_i, \mathbf{r}_i, \mathbf{o}_i; \mathcal{T}_M) > \max_{S_{\mathrm{M}} \in \mathcal{S}_M^E} \left\{ \frac{1}{n} \sum_{i=1}^n \mathbf{K}(\mathbf{s}_i, \mathbf{r}_i, \mathbf{o}_i; S_{\mathrm{M}}) \right\} + \Delta \tag{4}$$

## 2.3 Estimation

We now discuss best practices for effective estimation hidden knowledge, independently of the specific choices for our study (§3.2). To evaluate hidden knowledge, we must (1) select an *internal* scoring function $\mathcal{T}_M$, (2) estimate the set of *external* functions $\mathbf{S}_M^E$, (3) compute $\mathbf{K}$ for each scoring function, and (4) compare $\mathbf{K}(\cdot; \mathcal{T}_M)$ with $\mathbf{K}(\cdot; \mathbf{S}_M)$ for any external function.

**Scoring Functions.** For $\mathbf{S}_M^E$, a natural choice is $P_{\mathrm{M}}(a \mid q)$. On top of that, we can include normalizing $P_{\mathrm{M}}(a \mid q)$ by $a$'s length (Wang et al., 2023) and prompting $M$ to verify $a$'s correctness (Kadavath et al., 2022). $\mathcal{T}_M$ can exploit internal model structures, such as hidden states. A common choice is a probing classifier (Hupkes et al., 2018; Belinkov & Glass, 2019).

$\mathbf{K}$. To estimate $\mathbf{K}(\mathbf{s}, \mathbf{r}, \mathbf{o}; \mathbf{S}_M)$ we need to (1) estimate $\mathbf{Q}(\mathbf{s}, \mathbf{r})$, $\mathbf{A}(\mathbf{o})$, and $\tilde{\mathbf{A}}(\mathbf{o})$, and (2) score each $(\mathbf{q}, \mathbf{a}) \in \mathbf{Q}(\mathbf{s}, \mathbf{r}) \times \tilde{\mathbf{A}}(\mathbf{o})$ with $\mathbf{S}_M$. $\mathbf{Q}(\mathbf{s}, \mathbf{r})$ can be obtained by paraphrasing one question. For $\tilde{\mathbf{A}}(\mathbf{o})$, it is typically computationally infeasible to enumerate all entities from the relevant type. E.g., if $\mathbf{r}$ is "*married to*", there are approximately **8.2 billion** people worldwide. Uniformly sampling a subset of entities may lead to trivial negatives, answers that are clearly incorrect and easily ranked below correct ones. To increase the likelihood of *negative* answers that are hard for $M$, we can sample answers from $M$. However, this introduces a risk of failing to sample *correct* answers that are deemed unlikely by $M$, or even not sampling correct answers at all. To mitigate this, we can manually include $\mathbf{o}$ and its paraphrases. Lastly, to estimate $\mathbf{A}(\mathbf{o}) \subseteq \tilde{\mathbf{A}}(\mathbf{o})$, we must determine the correctness of each $\tilde{\mathbf{a}} \in \tilde{\mathbf{A}}(\mathbf{o})$ by comparing it to $\mathbf{o}$. Since string matching is sensitive to paraphrasing or differences in granularity (Yona et al., 2024), we recommend using an LLM-as-a-judge (Wei et al., 2024).

# 3 Study Design

We now present the design of our study that approximates hidden knowledge according to our definition (§2). In our main setup throughout the paper we use three popular open-

weights LLMs: Llama-3-8B-Instruct (Dubey et al., 2024), Mistral-7B-Instruct (Jiang et al., 2023), and Gemma-2-9B-Instruct (Team et al., 2024). To provide evidence of hidden knowledge in larger models, in §A.10 we also run the main experiment with Qwen3-32B (Yang et al., 2025), but we use a smaller set of answers to approximate $\tilde{\mathbf{A}}(\mathbf{o})$ (see §3.2) due to compute budget limitations. More details on the models are in §A.12.

### 3.1 Collecting the Set of Factual Triplets $\mathcal{D} = \{(\mathbf{s}_i, \mathbf{r}_i, \mathbf{o}_i)\}_{i=1}^n$

We build on EntityQuestions (Sciavolino et al., 2021), as it contains triplets from Wikidata (Vrandečić & Krötzsch, 2014) that were already converted into QA pairs. We categorize the relations (Table 2 in the appendix) to focus on ones that are (1) hard to guess and (2) have a single, unambiguous answer, making them easy to grade. Specifically, we use P26 (spouse), P176 (manufacturer), P264 (record label), and P50 (author). To form the development and test sets, we use 500 questions per-relation from the EntityQuestions test split, allocating 10% for development. The training set is based on 500 questions per-relation from the EntityQuestions train split. The training and development sets are used only to train the internal scoring function. Full details on the data creation process are in §A.1 and data statistics are in Table 3 in the appendix.

### 3.2 Approximating the Quantities of Interest

$\mathbf{S}_M^E$. There are two main paradigms for *external* scoring functions, which operate on the $M$'s observable token-level probabilities, we test both:

- **Production-oriented**: Measure how well $M$ assigns probability mass to $a$ when prompted to answer $q$, under its autoregressive decoding process. We compute this likelihood using $M$'s token-level probability, $\mathbf{P}(\mathbf{a}|\mathbf{q}) = \prod_{i=1}^n P(a_i \mid q, a_{<i})$, as well as its length-normalized variant (Brown et al., 2020; Wang et al., 2023): $\mathbf{P_{norm}}(\mathbf{a}|\mathbf{q})$ $= (\prod_{i=1}^n P(a_i \mid q, a_{<i}))^{\frac{1}{n}} = \exp\left(\frac{1}{n} \sum_{i=1}^n \log P(a_i \mid q, a_{<i})\right)$.

- **Verification-oriented**: Estimate the likelihood of $M$ to *classify* $a$ as a correct answer to $q$. This classification can be implemented either by prompting $M$ to generate verbal scores (Lin et al., 2022; Tian et al., 2023), or by prompting it to assess $a$'s correctness and inspecting the likelihood of generating the "True" token (Kadavath et al., 2022). We focus on the latter as it worked substantially better in early experiments.[4] Formally, we use $\mathbf{P}(\mathbf{True})$, defined as $P(\text{"True"} \mid q, a)$.

$\tilde{\mathbf{A}}(\mathbf{o})$. To approximate the set of plausible answers, we (1) generate one answer with greedy decoding and (2) sample additional $1,000$ answers with a temperature of 1. Figure 1 demonstrates the diminishing returns of sampling more answers. E.g., for $P50$, the last 200 answers (beyond the first 800) contributed 0.003 to the probability mass, indicating that answers obtained at this stage are either duplicates or have very low token-level probabilities. Next, (3) we add the gold answer from EntityQuestions (i.e., $\mathbf{o}$), when it was not sampled, which occurs in 64% of questions on average.

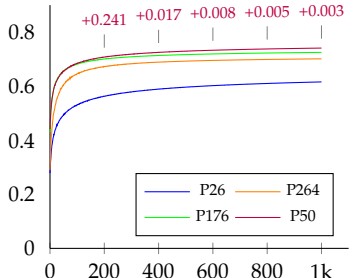

$\mathbf{A}(\mathbf{o})$. The set of correct answers is estimated using an LLM judge that compares each $\tilde{a} \in \tilde{\mathbf{A}}(\mathbf{o})$ to $\mathbf{o}$. We discard ∼8% of the questions where all sampled answers are correct, as they lack comparative value; alternatively, we could manually set $\mathbf{K}$ to 1. More details and a human evaluation of the judge are in §A.3.

Figure 1: Accumulated probability mass as a function of sample size for Mistral-7B-Instruct. For P50, the difference between every 200 samples is annotated.

$\mathbf{Q}(\mathbf{s}, \mathbf{r})$. We use the original question from EntityQuestions (i.e., $|\mathbf{Q}(\mathbf{s}, \mathbf{r})| = 1$), avoiding paraphrasing because our experiments are already computationally expensive: they require sampling a large number of answers per question, annotating each with an LLM judge, and scoring them using three different methods.

---

[4]We tested generating verbal scores on different scales (0-1, 1-10, 1-100), but the performance was poor. We note that verbal scores have mainly shown effectiveness in *reasoning* tasks.

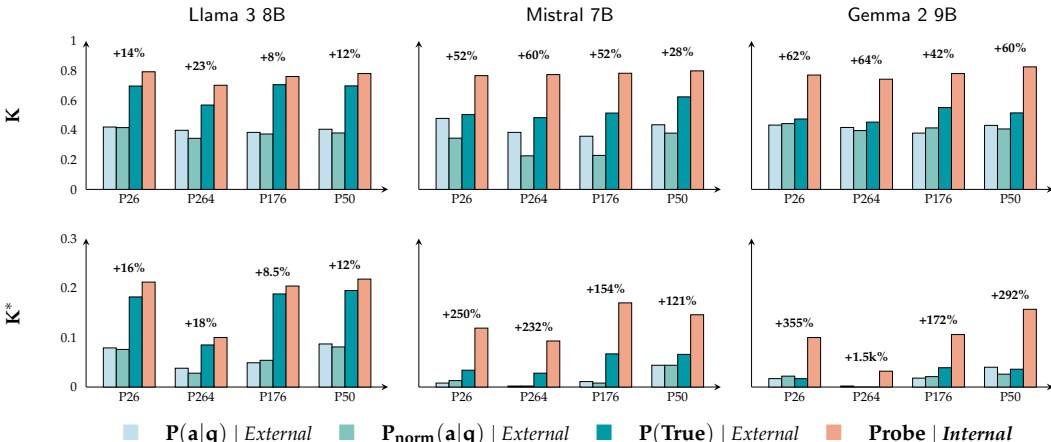

Figure 2: Average **K** (top) and **K**$^*$ (bottom) scores, as defined in equations 2 and 3, for each scoring function, relation and model. The bars are sorted according to the order in the legend (for color-blind readers), and the percentage difference between the best-performing external scoring function and our internal scoring function (**Probe**) is annotated. *All* those differences are statistically significant with $p < 0.05$.

**Δ**. We set the threshold Δ (Equation 4) dynamically per setup based on statistical significance by performing a paired t-test with a p-value of 0.05. Technical details are in §A.7.

$\mathcal{T}_M$. The space of possible internal functions is vast, and finding the optimal one is beyond this work's scope. Since our goal is to explore the *existence* of hidden knowledge, demonstrating it with a single internal function is sufficient. Consequently, our results should be interpreted as a lower bound. We pick $\mathcal{T}_M$ to be a probing classifier (Ettinger et al., 2016; Belinkov et al., 2017; Hupkes et al., 2018; Belinkov & Glass, 2019). Specifically, our probe is a linear classifier, trained with a logistic regression objective, that receives $M$'s hidden state $h_M(\mathbf{q}, \mathbf{a})$, obtained by encoding $q$ and $a$, and classifies $a$ as correct or incorrect. We then use its output probability, representing the likelihood that $a$ is correct, as $\mathcal{T}_M$. The probe is trained on $(q, a)$ pairs labeled for correctness, with full technical details provided in §A.5. We create the training data based on the training split using the process described in §A.1. We train probes for all layers and choose the best layer based on a development set.

The fact that the probe is trained introduces a risk that it may rely on knowledge acquired during its own training, rather than reflecting $M$'s internal representation of truthfulness. We mitigate this through careful data curation, ensuring that the factual information present in the training set is not useful for classifying test examples. Since we use QA pairs that map to *(subject, relation, object)* triplets, we can ensure that there are no subject and object overlaps between the training and test splits. In §A.6 we also empirically verify that the factual information from the training set is not useful for classifying test examples.

Lastly, we ensure that we train *mostly* on questions for which we have high certainty that $M$ knows the answer. The intuition behind this is that we want to train the probe to distinguish between multiple answers to the *same question*, so we want to ensure that when we train on a $(q, a)$ pair, $M$ encodes the information about $a$'s correctness. We refer to this approach as knowledge-aware probing and discuss it detail in §A.5.

## 4 Results

### 4.1 Evidence of Hidden Knowledge in LLMs

Figure 2 presents the average **K** and **K**$^*$ scores across all three models and four relations. Notably, across all 12 setups, the average **K** based on the *internal* scoring function is larger than based on any *external* one, with all the differences being statistically significant (see §A.7). This is also the case for the stricter definition of full knowledge **K**$^*$. This provides a strong evidence of the existence of hidden knowledge in LLMs. The magnitude of the

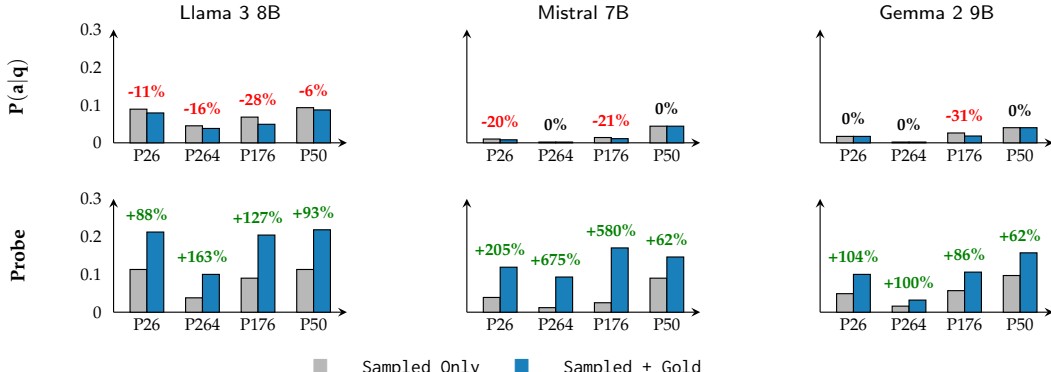

Figure 3: Comparison of average $\mathbf{K}^*$ values (see Equation (3)), under two conditions: *without* manually adding the gold (Sampled Only, left bars) and *with* manually adding the gold (Sampled + Gold, right bars).

gap between internal and external knowledge varies substantially across models, with an average difference in **K** of 57% for Gemma but only 14% for Llama. This indicates that models differ in their ability to fully expose their knowledge, likely due to factors such as differences in training data, methodologies or architecture, calling for future work to explore how to better facilitate knowledge exposure.

When examining the *external* scoring functions, we see a clear advantage for $\mathbf{P}(\mathbf{True})$, as it outperforms other external functions in every setting. $\mathbf{P}(\mathbf{True})$ also accounts for the relatively low magnitude of hidden knowledge in Llama, as the gap between **Probe** and the other two external functions is considerably larger. This shows that LLMs can exhibit a gap between the answers they can *generate*[5] with reasonable likelihood and those they can *verify* as correct when prompted to do so (Huang et al., 2025; Rodriguez et al., 2025).

## 4.2 LLMs Can Fail to Generate Facts They *Fully* Know, Even After 1,000 Attempts

In human cognition, the *tip of the tongue* state describes inability to recall a known word (Brown & McNeill, 1966). Psycholinguistics distinguishes *comprehension* from *production*, noting that people may understand words yet fail to retrieve them (Treiman et al., 2003). Linguistic theory emphasizes that *performance* (language use) may not reflect *competence* (language knowledge) (Chomsky, 1965). Could LLMs exhibit a similar cognitively plausible gap between retrieval mechanisms and their internal knowledge?

We have shown that LLMs encode more knowledge than they express externally. We now demonstrate that some knowledge is so deeply hidden that $M$ struggles to even consider a known correct answer as a candidate during generation. To this end, we take a closer look at the impact of manually adding the gold answer $a_G$ to $\tilde{\mathbf{A}}(\mathbf{o})$ when it was not sampled (§3.2). In this setup, $a_G$ is a *correct* answer that $M$ is highly *unlikely to generate*, as it was not sampled after 1k attempts, allowing us to test if $M$ can know $a_G$ despite failing to generate it. Figure 3 compares $\mathbf{K}^*$ scores between using only the sampled answers as $\tilde{\mathbf{A}}(\mathbf{o})$ and our main setup (§4.1) where $a_G$ is manually added. We focus on our notion of *full* knowledge $\mathbf{K}^*$ since we aim to show cases of perfect knowledge but failure to generate (we report and discuss **K** scores in §A.13).

As we would expect, adding $a_G$ never improves $\mathbf{K}^*$ scores for $\mathbf{P}(\mathbf{a}|\mathbf{q})$ (top row). Any improvement would require $\mathbf{K}^*$ to transition from 0 to 1 for certain questions, which is highly unlikely since it will require $a_G$ to receive a higher score than any incorrect candidate.

---

[5] Since autoregressive generation involves sampling from $M$'s token probability distribution, the likelihood assigned to $a$ given $q$ (measured by $\mathbf{P}(\mathbf{a}|\mathbf{q})$) directly reflects how likely $M$ is to generate it. Therefore, if $M$ assigns higher $\mathbf{P}(\mathbf{a}|\mathbf{q})$ scores to correct answers than to incorrect ones, it is more likely to generate a correct answer.

| $\mathbf{P(a|q)}$ | | $\mathbf{P_{norm}(a|q)}$ | | $\mathbf{P(True)}$ | | **Probe** | |
|---|---|---|---|---|---|---|---|
| Answer | Score | Answer | Score | Answer | Score | Answer | Score |
| BMW $^-$ | 0.761 | BMW $^-$ | 0.873 | BMW Group $^-$ | 0.980 | Volvo Buses $^+$ | 0.465 |
| Volvo $^+$ | 0.012 | BMW Group $^-$ | 0.114 | Volvo Buses $^+$ | 0.980 | Volvo $^+$ | 0.080 |
| BMW Group $^-$ | 0.001 | Volvo $^+$ | 0.110 | BMW $^-$ | 0.941 | BMW Group $^-$ | 0.065 |
| Stellantis $^-$ | $\approx 0$ | Stellantis $^-$ | 0.041 | Volvo $^+$ | 0.926 | BMW engines $^-$ | 0.028 |
| BMW engines $^-$ | $\approx 0$ | BMW engines $^-$ | 0.036 | BMW engines $^-$ | 0.245 | BMW $^-$ | 0.024 |
| Volvo Buses $^+$ | $\approx 0$ | Volvo Buses $^+$ | 0.001 | Stellantis $^-$ | $\approx 0$ | Stellantis $^-$ | 0.002 |
| $\mathbf{K} = 0.375\ (3/8)$ | | $\mathbf{K} = 0.25\ (2/8)$ | | $\mathbf{K} = 0.625\ (5/8)$ | | $\mathbf{K} = 1\ (8/8)$ | |

Figure 4: A real example of scores assigned to each answer $a \in \tilde{\mathbf{A}}(\mathbf{r})$ to the question *"Which company is Volvo B58 produced by?"*, according to each scoring function using Gemma 2 9B. Answers are sorted by score, with correct ones colored in green and marked $^+$, and incorrect ones colored in red and marked $^-$.

Notably, for **Probe** (bottom row), we observe considerable improvements across all setups, indicating that $\mathbf{K}^*$ often transitions from 0 to 1. This happens when there were no correct answers sampled, and thus $\mathbf{K}^*$ was manually set to 0, yet when $a_\text{G}$ was added to $\tilde{\mathbf{A}}(\mathbf{o})$, **Probe** ranked it higher than *all* the incorrect candidates. This demonstrates an extreme case of hidden knowledge: $M$ fails to generate $a_\text{G}$ after $1k$ attempts, yet still perfectly knows that $a_\text{G}$ is the correct answer, as it is able to rank it higher than any incorrect candidate.

To further substantiate this finding, we directly quantify cases where $M$ has perfect knowledge of a fact but is extremely unlikely to consider a correct answer during generation. We define such cases by the following conditions: (1) No correct answer was sampled after $1k$ attempts, (2) $\mathbf{P(a^G \mid q)} < \mathbf{0.01}$, and (3) $\mathbf{K}^* = 1$. On average, these cases occur in 7.2% of the questions, demonstrating that they are not just rare cases with negligible impact. This finding highlights a fundamental limitation in the generation process of LLMs. While it is expected that $M$ may occasionally generate an incorrect answer despite knowing the correct one (Simhi et al., 2024), it is highly surprising that a known correct answer is practically *never* generated, not even once, even with large-scale repeated sampling. This limitation holds to many popular decoding methods that use next-token probabilities, since they are guided by the $\mathbf{P(a|q)}$ distribution, meaning that if $\mathbf{P(a|q)}$ is low, the chance of generating **a** remains low. Understanding the cause of this limitation and how to mitigate it is an important direction for future research on decoding mechanisms. The solution may lie in decoding paradigms that consider internal signals, e.g., Rimsky et al. (2024).

### 4.3 A Case Study

Figure 4 presents a case study, comparing scores assigned to each answer $a \in \tilde{\mathbf{A}}(\mathbf{o})$ to the question *"Which company is **Volvo B58** produced by?"*. Since *"Volvo"* appears in the text, the question may seem easy. However, while *"Volvo B58"* refers to a bus made by *"Volvo Buses"*, the term *"B58"* is also the name of an engine produced by *"BMW"*. So the model must recognize that *"B58"* refers to a Volvo bus, not the BMW engine. It is particularly evident that the likelihood of *generating*[5] a correct answer, reflected by $\mathbf{P(a|q)}$ and $\mathbf{P_{norm}(a|q)}$, is extremely low. Interestingly, even though $\mathbf{P(True)}$ is also an *external* score, it ranks "Volvo Buses" significantly higher, demonstrating the gap discussed in §4.1 between the ability to generate and the ability to verify correctness. However, $\mathbf{P(True)}$ assigns the same score to "Volvo Buses" and the wrong answer "BMW Group", indicating that $M$ struggles to distinguish between them. Despite that, the *internal* scoring function perfectly ranks the answers, providing a real example of a case where $M$ encodes the knowledge in its parameters but fails to express it externally. Lastly, the gold answer in this case is "Volvo Buses", but it was not generated by $M$ in all 1,000 samples; only "Volvo" was sampled, possibly because *"Volvo"* appears in the question. This illustrates the limitations in LLMs' generation capabilities we discussed in §4.2.

This example also highlights the importance of our two key design choices. First, assessing knowledge w.r.t. to a large set of answers ensures a meaningful comparison. E.g., if we considered only "Volvo" and "BMW engines", all methods would yield a similar ranking. Second, approximating $\tilde{\mathbf{A}}(\mathbf{o})$ by sampling from $M$ is very useful as it allows to automatically generate challenging candidates like "BMW".

| | Llama-3-8B | Mistral-7B | Gemma-2-9b | Average |
|---|---|---|---|---|
| Greedy | 22.1 | 18.8 | 22.7 | 21.2 |
| Random | 16.4* (-25.8%) | 12.3* (-34.6%) | 11.3* (-50.2%) | 13.3* (-36.9%) |
| Majority | 23.7* (+7.2%) | 19.7* (+4.8%) | 22.6 (-0.4%) | 22.0* (+3.9%) |
| $\mathbf{P(a|q)}$ | 23.6* (+6.8%) | 20.0* (+6.4%) | 23.2* (+2.2%) | 22.3* (+5.1%) |
| **Probe** | **25.4*** (+14.9%) | **22.0*** (+17.0%) | **23.7*** (+4.4%) | **23.7*** (+12.1%) |
| Oracle | 44.2* (+100.0%) | 38.9* (+106.9%) | 49.8* (+119.4%) | 44.3* (+108.8%) |
| **Probe** w. gold | 34.5* (+56.1%) | 33.9* (+80.3%) | 27.6* (+21.6%) | 32.0* (+52.7%) |

Table 1: Closed-book QA accuracy when selecting the top-scoring answer from 1k samples using different scoring methods. Oracle score each correct answer as 1 and each wrong one as 0. **Probe** w. gold includes the gold answer if it was not sampled. * marks statistically significant differences ($p < 0.05$) from greedy.

### 4.4 Increasing Test-Time Compute via Repeated Answer Sampling and Ranking in Closed-Book QA

A practical implication of hidden knowledge is the potential to develop methods that better expose it, improving downstream task performance. A simple approach is to sample multiple answer candidates and select the correct one (Brown et al., 2024; Hassid et al., 2024; Zhao et al., 2025).[6] We test its feasibility in our setup when sampling 1k answer candidates and selecting the highest-scoring one, aiming to surpass greedy decoding. Table 1 presents the results. For brevity, we aggregate all relations per model.

Notably, greedy decoding performs poorly, indicating a challenging setup, likely due to long-tail knowledge requirements. Interestingly, even with *short* entity-focused answers, greedy decoding does not always select the globally optimal path, as evidenced by the improved performance when selecting answers using $\mathbf{P(a|q)}$. This highlights the importance of the $\mathbf{P(a|q)}$ baseline in controlling for a major confounder: ensuring that the probe does not simply resort to selecting the answer with the highest global token-level probability. **Probe** demonstrates notable relative improvements of 12% on average across LLMs, providing further evidence of the potential of self-verification based on the model's hidden states to enhance downstream performance (Orgad et al., 2025). The Oracle baseline, where a correct sampled answer is always ranked first, provides an upper bound for a perfect scoring method. The fact that it remains the highest, shows that for a considerable amount of questions, we successfully sample a correct answer but fail to identify it, which could be attributed to guessing rather than knowing (Yona et al., 2025). However, the most interesting results are for **Probe** w. gold, where the gold answer $a_G$ is manually included if not sampled. Consistent with §4.2, $a_G$ would often be selected as the top answer if only it was sampled, which could lead to 52% average improvement over greedy (i.e. additional 40% compared to **Probe**). This result highlights a substantial potential for improving performance that remains *inaccessible* due to the constraints we discovered in the LLMs' generation capabilities. A natural step for future work is to develop better sampling methods that produce high-quality and diverse candidates to boost test-time performance.

## 5 Related Work

**Knowledge of LLMs.** The factual knowledge of LLMs has been widely studied. Early work considered a model to know a fact if it correctly completed a cloze sentence (Petroni et al., 2019; Jiang et al., 2020; Kassner et al., 2020, *inter alia*), or directly answered a question, either in a zero-shot setting (Radford et al., 2019) or after fine-tuning (Roberts et al., 2020). Modern LLMs, capable of instruction following, are typically directly prompted to answer questions (Wei et al., 2024; Singhal et al., 2023; Anil et al., 2023; Dubey et al., 2024; Cohen et al., 2023, *inter alia*). These efforts have largely been guided by what appears intuitively reasonable, without a clear definition of *knowledge* (Fierro et al., 2024). Beyond the lack of a formal definition, a key limitation in previous work is that even though studies emphasized the importance of evaluating predictions across semantically equivalent *questions* (Elazar

---

[6]It is important to stress that while higher $\mathbf{K}$ increases the chances of success under inference scaling, it does not guarantee it. In §A.14, we include a detailed discussion on the relationship between our $\mathbf{K}$ measure and inference scaling.

et al., 2021; De Cao et al., 2021; Zheng et al., 2023), most of them focused on evaluating a single model *response*. As discussed in §2.1, we posit that the relative ranking of all plausible answers is important and design our definition to reflect this. One aspect we leave out of scope is verifying related facts when measuring knowledge (Kassner et al., 2021; Zhong et al., 2023; Cohen et al., 2024). E.g., to conclude that a model knows that Paris is France's capital, we may also check that it knows that Paris is a city in France. We hope future research will explore corresponding extensions to our definition.

**Hidden Knowledge in LLMs.** Findings from prior work suggest that LLMs *may* encode what we define as hidden knowledge. There is a growing evidence that LLMs encode *truthfulness* information, enabling assessment of *individual* candidate answers' correctness either via probing the model's *internal* states (Burns et al., 2023; Azaria & Mitchell, 2023; Marks & Tegmark, 2024, *inter alia*) or by prompting it directly (Lin et al., 2022; Kadavath et al., 2022; Tian et al., 2023), with these approaches not always agreeing (Liu et al., 2023). Other studies showed that a model can be "steered" to answer correctly where it previously failed (Li et al., 2023b; Zhang et al., 2024; Tulchinskii et al., 2024; Rimsky et al., 2024). Turpin et al. (2023) showed a model can be biased by its input structure towards incorrect answers about known facts, and even generate a plausible justification. Gekhman et al. (2024) showed that fine-tuning on new knowledge can cause hallucinations on facts that were known to the pre-trained model, raising questions about whether those facts are fully forgotten. Indeed, some results suggest that even when a fine-tuned model fails to recall a fact, it may still encode information about it in its representations (Gottesman & Geva, 2024; Patil et al., 2024). These findings only hint at the existence of hidden knowledge but do not clearly define or systematically demonstrate it. For example, studies that show that LLMs encode truthfulness information usually test the ability to classify *individual* statements as correct or incorrect. However, success in such classification could stem from uncertainty representation rather than factual knowledge. E.g., knowing that an answer is wrong does not guarantee knowledge of the right answer. Although our probe is trained in a related manner, we differ since rather than evaluating performance across a large set of individual QA pairs, we quantify the ability to correctly rank all possible answer candidates *for a specific question*, which is much more likely to reflect knowledge of the relevant fact.

**Scaling Test-Time Compute.** Considerable progress has been made in improving performance by increasing inference compute (Snell et al., 2025; OpenAI, 2024; Guo et al., 2025). A popular approach is to sample diverse responses and use verifiers to identify the correct one (Brown et al., 2024; Hassid et al., 2024; Zhao et al., 2025). Most studies focus on *reasoning* tasks, raising the question of whether such approaches can be effective for knowledge-intensive QA with *short answers*, where reasoning diversity matters less. Orgad et al. (2025) provided initial evidence by using a probing classifier to select the best answer among 30 model-generated candidates. We extend this evidence in a highly controlled setup facilitated by our framework and, more importantly, show that further performance gains are possible but constrained by the model's generation capabilities, specifically its ability to recognize the correct answer while failing to generate it as a candidate.

## 6 Conclusion and Future Work

We present a framework for assessing the extent to which a model encodes more factual knowledge in its parameters than it expresses in its outputs. Our definition of *knowledge* addresses limitations of measuring it based on a single generation's performance, and enables a unified measure of both *internal* and *external* knowledge, facilitating our definition of *hidden knowledge*. We then conduct a controlled study which indicate that LLMs consistently exhibit hidden knowledge to varying degrees, stressing the need to understand these differences and build models that better use their knowledge, for which our framework can serve as a foundation. We provide an extended discussion about potential reasons for hidden knowledge and possible mitigations in §A.16. We also demonstrate an extreme case of hidden knowledge, where the model perfectly knows an answer but is highly unlikely to generate it even once via repeated sampling. This highlights a limitation in the LLMs' generation capabilities which puts a practical constraint on scaling test-time compute via repeated answer sampling in closed-book QA, and opens an interesting direction for future research on decoding mechanisms.

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

# A Appendix

## A.1 Data Creation Process

As discussed in §3.1, we build on EntityQuestions (Sciavolino et al., 2021), which provides triplets from Wikidata (Vrandečić & Krötzsch, 2014) that have already been converted into QA pairs. We categorize relations based on two criteria: (1) they should be difficult to guess, and (2) they should have a single, unambiguous answer, making them easier to grade. This categorization is presented in Table 2.

We consider a question easy to guess when the object's entity type has a relatively small set of unique values, and when the same value can apply to multiple subjects for a given relation, making it possible to guess a more prevalent value with reasonable success. For example, guessing a capital city is easier than guessing a spouse since there are only about 200 possible capitals, compared to billions of potential individuals. Similarly, professions can be easier to guess because certain professions are more common, whereas each spouse relationship is unique to a single individual.

We define a relation as well-defined if it has a single, specific answer. In contrast, some relations allow multiple levels of granularity, making them ambiguous. For instance, when specifying a location, responses may vary in detail (e.g., city, state, or country), making it less well-defined.

We use P26 (spouse), P176 (manufacturer), P264 (record label), and P50 (author). While P40 (child) could also be included, answering it often relies on knowledge closely related to P26 (spouse). Given the computational expense of our experiments, we decided to exclude it to manage resources more effectively.

**Test and Dev Sets.**   We use the test split of `EntityQuestions` for each of our four relations. We filter out questions with more than one gold answer and questions that contain the gold answer. We also deduplicated questions that appeared more than once. We then sample 500 questions from each of our four relations. Afterwards, we generate the greedy answer and sample additional 1,000 answers with a temperature of 1. To this end we prompt the model with an instruction to answer the question, as described in §A.2. Next, we label each of the sampled answers using the LLM judge as described in §A.3. Some answers get labeling errors (see §A.3) in which case we filter the whole question out. We considered filtering the answers but ultimately chose to filter entire questions instead, as answer-level filtering could introduce noise into our evaluation. Our sampled answers are intended to approximate the full set of plausible responses, and we aim to utilize this entire set when estimating knowledge. Finally, we reserve 10% of the remaining questions for a developments set. Full statistics can be found in Table 3.

**Train set.**   We use the train split of `EntityQuestions` for each of our four relations and apply the same filtering steps described above (for the test and dev sets). To ensure that there is no knowledge leakage between the train and test sets we also apply the following filtering steps: (1) Filter questions that appear in the test set. (2) For relations where the subject and object have the same entity type, e.g., P26 ("married to") we ensure that the train subject does not appear as object in the test and vice versa.

We focus on questions for which the greedy answer is correct (see §A.4). To do this, we first predict greedy answers for all examples and retain only those with an exact match to the gold answer. Using exact match allows us to avoid running the judge across the full test set, which is large. This approach is justified as our goal is to find a sufficient number of examples where the greedy answer is correct, not to exhaustively identify all such cases. For these selected examples, we treat the greedy answer as the positive case and sample 200 additional responses using a temperature of 2.[7] We label the sampled answers using our LLM judge and discard questions for which no incorrect answers were generated. For the remaining questions, we randomly select one incorrect answer along with the correct greedy answer for our final dataset. From this set, we randomly sample 500 questions per relation, resulting in a total of 2,000 examples.

## A.2   QA Prompts

This section describes the prompt that we used to sample answers from our evaluated LLMs. In designing our prompts, our goal was to instruct the model to generate plausible answers without unnecessary descriptive words while keeping the instruction natural. We did not specifically optimize the prompt instructions. It is possible that a carefully crafted

---

[7]A higher temperature increases the likelihood of sampling incorrect answers, which can be difficult when the greedy response is already correct.

| Relation | Question Template | Hard to Guess? | Well Defined? |
|---|---|---|---|
| P176 | Which company is [X] produced by? | ✓ | ✓ |
| P264 | What music label is [X] represented by? | ✓ | ✓ |
| P50 | Who is the author of [X]? | ✓ | ✓ |
| P26 | Who is [X] married to? | ✓ | ✓ |
| P40 | Who is [X]'s child? | ✓ | ✓ |
| P106 | What kind of work does [X] do? | ✗ | ✓ |
| P112 | Who founded [X]? | ✓ | ✗ |
| P127 | Who owns [X]? | ✓ | ✗ |
| P131 | Where is [X] located? | ✓ | ✗ |
| P136 | What type of music does [X] play? | ✗ | ✓ |
| P159 | Where is the headquarter of [X]? | ✓ | ✗ |
| P17 | Which country is [X] located in? | ✗ | ✓ |
| P170 | Who was [X] created by? | ✓ | ✗ |
| P175 | Who performed [X]? | ✓ | ✗ |
| P19 | Where was [X] born? | ✓ | ✗ |
| P20 | Where did [X] die? | ✓ | ✗ |
| P276 | Where is [X] located? | ✓ | ✗ |
| P36 | What is the capital of [X]? | ✗ | ✓ |
| P407 | Which language was [X] written in? | ✗ | ✓ |
| P413 | What position does [X] play? | ✗ | ✓ |
| P495 | Which country was [X] created in? | ✗ | ✓ |
| P69 | Where was [X] educated? | ✓ | ✗ |
| P740 | Where was [X] founded? | ✓ | ✗ |
| P800 | What is [X] famous for? | ✓ | ✗ |

Table 2: Overview of the relations, their corresponding question templates, and metadata about difficulty and entity type definition. *"Hard to Guess"* refers to questions where the possible answer's space is large. For instance, person names are considered hard to guess, whereas professions are not, as there are relatively few professions, and the model can default to more common ones. *"Well Defined"* assesses whether the entity type and answer granularity are unambiguous. For example, location can refer to a city, country, or exact address, making it less well-defined. Similarly, ownership may refer to a person or a corporation, adding ambiguity. In contrast, capital city or company name are well-defined entity types with a clear level of granularity.

| | Llama-3-8B | | | Mistral-7B | | | Gemma-2-9B | | |
|---|---|---|---|---|---|---|---|---|---|
| | Test | Dev | Train | Test | Dev | Train | Test | Dev | Train |
| P26 | 445 | 50 | 500 | 400 | 45 | 500 | 425 | 48 | 500 |
| P264 | 447 | 50 | 500 | 430 | 48 | 500 | 442 | 50 | 500 |
| P176 | 430 | 48 | 500 | 412 | 46 | 500 | 421 | 47 | 500 |
| P50 | 448 | 50 | 500 | 441 | 50 | 500 | 447 | 50 | 500 |
| Total | 1770 | 198 | 2000 | 1683 | 189 | 2000 | 1735 | 195 | 2000 |

Table 3: Dataset statistics across different models.

instruction could yield significantly better performance. Yet our goal is to sample plausible answers from the model. We iterated through several versions until we observed that the model consistently outputs entities of the correct type (e.g., a person for "married to"). Once we reached this point, we did not observe meaningful variance in performance. The following system and user prompts were used for Llama and Mistral.

System Prompt (Llama and Mistral):

> Your job is to answer an entity-centric question.
>
> You need to answer with the correct entity, without any additional information.

User Prompt (Llama and Mistral):

> Here is the question. Simply reply with the correct entity. If you cannot answer for any reason, output None. But do try your best to find the correct answer.

```
`` `
```

Question: {question}

```
`` `
```

Just return the answer, with no text around it.

Gemma does not support a system prompt, and concatenating the system and user prompts from above did not work well enough. We ultimately used the following prompt.

User Prompt (Gemma):

Answer the following entity-centric question, reply with the correct entity without any additional information.

```
`` `
```

Question: {question}

```
`` `
```

Just return the answer, with no text around it.

### A.3  LLM Judge

We select Qwen2.5 14B Instruct (Yang et al., 2024)[8] as the grader for two reasons. First, it belongs to a different model family than the ones we evaluated. Second, it is larger. Larger models contain more factual knowledge, which can be useful for the judge task. In early experiments, we found that the performance of both zero-shot and *vanilla* chain-of-thought prompting was insufficient. Therefore, we designed a *program-guided chain-of-thought* prompt. Essentially, we construct a mini decision tree and prompt the LLM to follow specific steps until it reaches a verdict. Our program also includes a self-verification step, which may result in an error label. For a very small number of questions we found an issue with the gold answer, so we also let the judge verify that the gold answer has correct entity type. We design the prompt per-relation with small adaptations to account for specifics of the entity type and the question format. Below is an example of our prompt for P26 ("married to"):

I will give you a question about the spouse of a person (e.g., "Who is Umberto I of Italy married to?"), a gold answer, and a proposed answer. You need to compare the proposed answer to the gold answer and assign it one of the possible grades using the steps below.

Possible grades are:

A: CORRECT

B: INCORRECT

C: WRONG_GOLD

D: ERROR

Spelling errors, synonyms, abbreviations, or hedging (e.g., "it is possible that") should not alter the grade if the person referred to in the proposed answer matches the gold answer.

The steps are:

Step 1: If the gold answer does not refer to a person, output "C" and finish. Otherwise, proceed to Step 2.

Step 2: If the proposed answer does not refer to a person, output "B" and finish. Otherwise, proceed to Step 3.

Step 3: If the proposed answer refers to the exact same person as the gold answer, output "A" and finish. Otherwise, proceed to Step 4.

---

[8] https://huggingface.co/Qwen/Qwen2.5-14B-Instruct

Step 4: Double check that both answers reflect a person and the proposed answer refers to a different person from the gold answer. If it does, output "B". Otherwise, output "D" and finish.

```
```

Question: {question}

Gold answer: {gold_answer}

Proposed answer: {answer}

```
```

Output your thinking steps. After that, finish your response with "Output:" and the letter (A or B or C or D). Do not provide any explanations.

To save inference calls we run the judge only when exact match is false. In certain instances, we observed that while the judge executes the program correctly and reaches the intended conclusion, it nonetheless produces an incorrect output. Those cases could be addressed by applying simple heuristics that checked "step 4". For instance if the output is "A" (correct), then step 4 should not contain the word "different", or if the output is "B" (wrong) then it should not contain "refer to the same entity". Eventually, this procedure lead to a very high labeling quality as we demonstrate in our human evaluation next.

|  |  | Accuracy | F1 | Precision | Recall |
|---|---|---|---|---|---|
| LLM | P26 | 99.96 | 98.16 | 96.39 | 100.00 |
|  | P50 | 99.92 | 96.93 | 94.04 | 100.00 |
|  | P176 | 99.76 | 95.16 | 90.76 | 100.00 |
|  | P264 | 100.00 | 100.00 | 100.00 | 100.00 |
| EM | P26 | 99.12 | 32.66 | 100.00 | 19.52 |
|  | P50 | 99.20 | 51.86 | 100.00 | 35.01 |
|  | P176 | 98.11 | 31.32 | 100.00 | 18.56 |
|  | P264 | 99.42 | 56.72 | 100.00 | 39.58 |

Table 4: Judge Performance Comparison: LLM vs. Exact Match (EM). The score values represent an average measured across all predictive models ($M$) per each relation.

**Estimating Judge Quality.** To validate the reliability of our findings, we perform a human evaluation for the performance of our LLM judge. The judge model, denoted by $\mathbf{J}$, determines whether a predicted answer $\mathbf{a}$, for a factual question $\mathbf{q}$, provided by a predictive model $\mathbf{M}$ is equivalent in meaning to a gold answer $\mathbf{g}$. Authors of the paper manually annotated 1,080 examples for correctness. Each example consists of the triplet $(\mathbf{q}, \mathbf{a}, \mathbf{g})$, along with $\mathbf{v}_J$, a predictive verdict made by $\mathbf{J}$.

We categorize all evaluation examples into three distinct groups based on how closely $\mathbf{a}$ matches $\mathbf{g}$ and the verdict provided by the judge:

- **Group 1 (Exact Match):** The predicted answer $\mathbf{a}$ exactly matches the gold answer $\mathbf{g}$.[9] All these cases are automatically labeled as correct (true positives), and are not part of the 1080 examples we manually annotate.
- **Group 2 (Judge-Positive, Non-exact match):** The judge verdicts these answers as correct even though they do not exactly match the gold answer $\mathbf{g}$.
- **Group 3 (Judge-Negative, Non-exact match):** The judge verdicts these answers as incorrect, and they do not exactly match the gold answer $\mathbf{g}$.

Due to significant class imbalance – with incorrect predictions by $\mathbf{M}$ vastly outnumbering correct ones (approximately 60:1) – random sampling would yield insufficient representations of examples with correct predicted answers. E.g., if we randomly annotate, 1000

---

[9]We run a normalization process on $\mathbf{a}$ before comparing it to $\mathbf{g}$.

examples, we likely get ~16 correct answers. To address this, we employed the following sampling approach:

- All examples from **Group 1** (exact matches) were automatically considered true positives, and also not included in our annotated set of $1,080$ examples.

- For **Groups 2 and 3**, we sampled an equal number of cases, 540 from each group. This 540 examples are evenly distributed among the three predictive models and four relations (i.e., 45 examples per model and relation).

All selected examples were manually labeled for correctness. However, because our sampling strategy does not reflect the actual distribution of classes, we applied a re-weighting step after annotation to accurately represent the true proportions of Groups 2 and 3 in the complete dataset. Table 5 illustrates the sampling and re-weighting clearly for relation P26:

| Group | Sampled Size per relation | Actual Dataset Size | Est. correct $\mathbf{v}_J$ | Est. incorrect $\mathbf{v}_J$ |
|---|---|---|---|---|
| 1 (Exact Match) | 669 | 669 | 669 | 0 |
| 2 (Judge-positive) | 135 | 2887 | 2757.1 | 129.9 |
| 3 (Judge-negative) | 135 | 311324 | 311324 | 0 |

Table 5: An example of sampling and re-weighting approach for judge quality evaluation for relation P26.

Consider the relation *P26* as an example (Table 5). Our dataset has 669 examples in Group 1 (exact matches), 2887 in Group 2 (judge-positive, non-exact match), and 311324 in Group 3 (judge-negative, non-exact match). All examples from Group 1 are true positives. To estimate true positives from Group 2, we manually annotated a sampled subset of 135 examples, finding 129 (95.5%) correct. Thus, we estimate that $0.955 \times 2887 = 2757.1$ additional examples in Group 2 are true positives. Combining these, we have an estimated total of $669 + 2757.1 = 3426.1$ true positives. Following this approach, we estimate the amount false-positives and false negatives. We then compute precision, recall, accuracy, and F1-score from these re-weighted counts for each relation and report these scores.

We present the results in Table 4. Notably, the judge achieves very high accuracy of more than 99%. This is expected as most cases are straightforward in nature, as reflected by the high accuracy of the exact match alternative. Nevertheless, we look into the non-straightforward cases, where determining correctness is less obvious. To quantify the benefits of our LLM-based approach, we compare its precision and recall to the widely-used exact-match metric. Exact-match has a precision of 100% by definition, but it may suffer from low recall when it classifies paraphrases of the correct answer as incorrect. As seen in Table 4, our LLM judge successfully identifies many correct answers missed by the EM judge, achieving a notably higher recall[10] while maintaining a very high precision. This improvement can be attributed to multiple valid formulations of a correct answer that EM fails to capture. Taken together, these result provide evidence of a high performance of our labeling mechanism and support the validity of our primary findings.

| | $M$ **knows the** answer to $q$ | $M$ **doesn't know** the answer to $q$ |
|---|---|---|
| *a* **is correct** | **(A)** Known and Correct | **(C)** Unknown and Correct |
| *a* **is wrong** | **(B)** Known and Wrong | **(D)** Unknown and Wrong |

Table 6: All possible conditions for a given question $q$ and candidate answer $a$. In knowledge-aware probing we train exclusively on categories (A) and (B) to ensure that the model ($M$) knows that the correct answer is correct and the wrong answer is wrong.

---

[10]We note that true recall is likely slightly below 100%. Yet our human evaluation strongly suggests that it is close to 100%.

### A.4 Knowledge-aware Probe (on $M$'s hidden states)

In this section, we provide an extended discussion of our intuition for training the probe on questions for which it mostly knows the answers. We primarily describe *possible* risks associated with alternative choices but did not empirically validate whether these risks manifest in our specific setup. This does not affect our conclusions since, as discussed in §3.2, our goal is to explore the existence of hidden knowledge, and demonstrating it with a single internal function is sufficient. Further investigation of this aspect could be an interesting direction for future work.

The main intuition is that we want ensure that we train mostly on questions for which we have high certainty that $M$ knows the answer. Since we want to train the probe to distinguish between multiple answers to the same question, we want to ensure that when we train on a $(q, a)$ pair, $M$ encodes the information about $a$'s correctness. If $M$ does not know the answer to $q$, its representations are unlikely to contain useful discriminative information about the correctness of different answers, and thus our trained probe may be less effective.

To train a probing classifier for $\mathcal{T}_M$, we need a dataset of $(q, a)$ pairs labeled for correctness. Prior work follows two main approaches to create such a dataset. Approach (i) is to pair each question $q$ with its gold answer $a_G$ for positive examples and using *fabricated* answers as negatives (Marks & Tegmark, 2024; Azaria & Mitchell, 2023; Su et al., 2024; Rateike et al., 2023; Li et al., 2023a). Since fabricated negatives are often unlikely according to $M$, this risks the probe learning $M$'s likelihood rather than correctness. To disentangle likelihood from truthfulness, we instead use plausible (model-generated) incorrect answers. Approach (ii) is prompting $M$ to generate answers and labeling correct and incorrect responses as positive and negative, respectively (Zou et al., 2023; Orgad et al., 2025; Snyder et al., 2024; Yüksekgönül et al., 2024). To illustrate the risk associated with it, we note that for a given question $q$ and a candidate answer $a$, we can examine two key aspects: (1) does $M$ know the answer to $q$? and (2) is $a$ a correct answer to $q$? Table 6 categorizes the outcomes of all possible responses to these questions. Since there is considerable correlation between the correctness of the generated answer and the model's knowledge, in (ii) we are likely training the probe mostly on categories **(A)** and **(D)**. This may train the probe to identify whether $M$ knows an answer rather than assessing the correctness of specific answers, weakening its ability to distinguish between answers to the same question. Instead, we introduce *knowledge-aware probing*, focusing on categories **(A)** and **(B)**. To approximate data from these categories, we make a relaxation and assume that if $M$ generates the correct answer via greedy decoding, it likely knows the answer. Thus, we focus exclusively on questions for which greedy decoding produces a correct answer. We then use this (correct) greedy answer as a positive example (A). To obtain a negative example (B), we induce a hallucinated response from $M$, even though it likely knows the correct answer (Simhi et al., 2024), by sampling additional responses at high temperature until an incorrect answer is found.

### A.5 Training the Probe (on $M$'s hidden states)

As described in §A.1, we create a training set of 2,000 questions, 500 from each relation, applying multiple filtering steps to ensure no factual knowledge overlaps with the test set. We then merge these datasets and use the resulting data to train our probe. In early experiments, we also trained a separate probe for each relation, but since the conclusions were similar, we opted for a simpler approach using a single probe. We also examined the effect of data size. We found that the probe's performance with 250 questions per relation (1,000 total) was very close to that with 500 (2,000 total), ensuring the probe is not under-trained. The probe is trained with a logistic regression objective, receiving as input $M$'s hidden state $h_M(q, a)$, obtained by encoding $q$ and $a$, and classifying $a$ as correct or incorrect. To represent $a$ relative to $q$, we follow a similar procedure to that described in §A.8.1, simulating a sequence where $M$ generates $a$ when prompted with $q$. In early experiments, we also tested the sequence from §A.8.2, where the model is prompted to verify $a$ as the answer to $q$, but the classifier's performance was similar in both cases. Finally, we use the probe's output probability – representing the likelihood that $a$ is correct – as

$\mathcal{T}_M$. We train one probe per layer and select the layer with the best performance on the development set. We also present the per-layer performance in Figure 5. We observe that scores tend to improve and then mostly stabilize in the final two-thirds of the network, typically starting around layers 11–12 out of 32. This suggests that the upper layers encode higher-quality and more consistent representations compared to earlier layers. Interestingly, we also see that there is often a tiny drop in the last layers, which are used during decoding. However, this drop is relatively small. Apart from that, we did not observe any interesting insights from analyzing the different layers. However, it was not our focus during the research, so it is possible that we missed some interesting insights there which could be explored in future work.

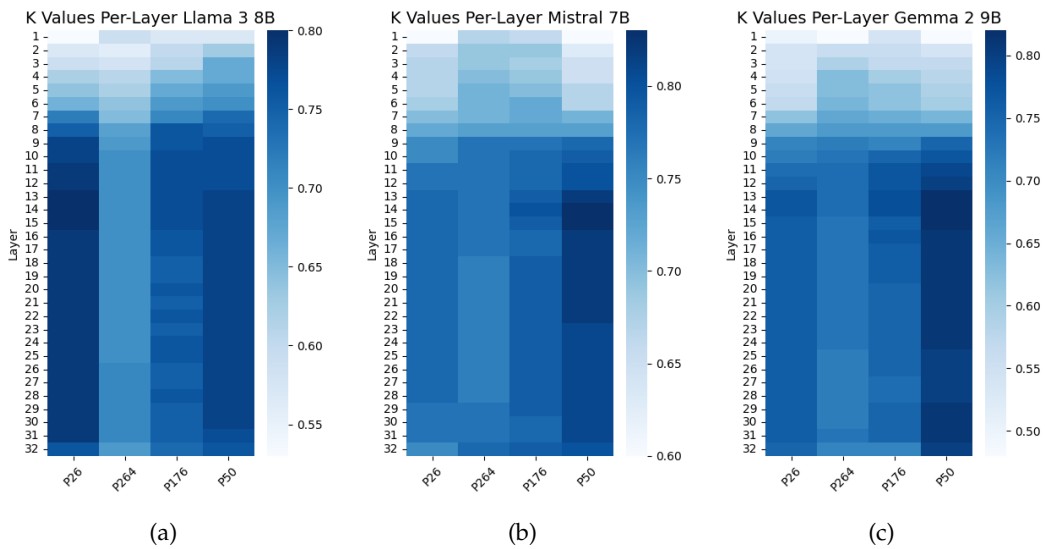

Figure 5: Per-layer **K** scores for Llama-3-8B-Instruct (a), Mistral-7B-Instruct (b) and Gemma-2-9B-Instruct (c).

## A.6   Evaluating Memorization in the Probe

As we discuss in §3.2 and §A.1, we ensure that the factual information present in the training set is not useful for classifying test examples by careful data curation. We now complement this by empirically verifying that the probing classifier's performance does not result from memorizing training examples. To this end, we compared the Probe's performance to baselines trained on alternative input representations.

We define $f(q, a)$ as the concatenation of a question $q$ and its corresponding answer $a$. In our probing setup, a classifier is trained on a hidden representation $h(f(q, a))$ extracted from the LLM.

As baselines, we trained classifiers using two alternative input representations:

1. **TF-IDF** features derived directly from the input text, denoted as $\texttt{TFIDF}(f(q, a))$.
2. **Embedding-based** features obtained by mean-pooling the token embeddings from the LLM's input embedding layer, denoted as $\texttt{EMBED\_MEAN}(f(q, a))$.[11]

For each of these representations, we trained a logistic regression classifier and two fully connected neural networks with varying depths. The idea is simple, if the training set contains facts that are useful for answering test questions, then we should see that leaning from these examples results in a better than random performance on the test set.

---

[11]This specific experiment was done with Qwen3-32B, to leverage the strongest embedding among all our LLMs.

Table 7 summarizes the accuracy results. Classifiers trained on both TF-IDF and embedding-based representations perform around chance level on the test set.[12] If the train set would contain useful information, we would expect at least minor gains on the test set. In contrast, the probe classifier, trained on internal LLM representations, significantly outperforms these baselines. This strongly suggests that the probe's performance is driven by genuine internal knowledge representations rather than memorization of textual content from the training examples.

| Input Representation | Model | Train Acc (%) | Test Acc (%) |
|---|---|---|---|
| - | Random | - | 50.0 |
| TFIDF($f(q,a)$) | Logistic Regression | 95.6 | 50.5 |
| TFIDF($f(q,a)$) | MLP (256, 256) | 99.9 | 50.7 |
| TFIDF($f(q,a)$) | MLP (512, 512, 256, 128) | 99.9 | 50.2 |
| EMBED_MEAN($f(q,a)$) | Logistic Regression | 98.8 | 49.8 |
| EMBED_MEAN($f(q,a)$) | MLP (256, 256) | 99.9 | 49.1 |
| EMBED_MEAN($f(q,a)$) | MLP (512, 512, 256, 128) | 99.9 | 50.9 |
| $h(f(q,a))$ | Logistic Regression | 99.9 | 64.0 |

Table 7: Accuracy of classifiers trained on different representations of the input $(q, a)$ pairs.

### A.7 Statistical Significance

In Figure 2, we report the outcomes of statistical significance tests comparing the **K** and **K**$^*$ values obtained using our internal scoring method against the best-performing external method for each model-relation combination. To this end, we shuffle all examples in the relevant test set, split them into 50 approximately equal-sized subsets, and compute **K** and **K**$^*$ for each subset. We then apply a paired-sample t-test with $p < 0.05$. We also report statistical significance in Table 1, where we compare each answer selection method to the greedy decoding baseline. The same procedure is applied, but we use 200 bins since we mix all four relations together.

### A.8 External Scoring

#### A.8.1 $\mathbf{P(a|q)}$ *and* $\mathbf{P_{norm}(a|q)}$

To compute $\mathbf{P(a|q)} = \prod_{i=1}^{n} P(a_i \mid q, a_{<i})$, as well as its length-normalized variant $\mathbf{P_{norm}(a|q)}$ $= (\prod_{i=1}^{n} P(a_i \mid q, a_{<i}))^{\frac{1}{n}} = \exp\left(\frac{1}{n}\sum_{i=1}^{n} \log P(a_i \mid q, a_{<i})\right)$ we need to obtain the token-level probabilities of the answer $a$ conditioned on the question: $\{P(a_i \mid q, a_{<i}) \mid i = 1,\ldots,n\}$. For each answer $a$, we use the relevant prompt from §A.2 as $q$ and construct a sequence $S$ that simulates the generation of $a$ given $q$. Instead of simply concatenating $a$ to $q$, we ensure that all special tokens match their expected form as if the model had actually generated $a$ following $q$. We then perform a forward pass through the model with $S$ as input and use the resulting logits to compute the token-level likelihoods of $a$. This procedure is particularly useful for scoring the gold answer in cases where the model did not generate it at all.

#### A.8.2 P(True)

We use the system and user prompts from below for Llama and Mistral, and concatenate them for Gemma. We do a forward pass with the model on the resulted input and then use the logit of the next token to compute the likelihood of "A". Rather than applying the softmax over the entire vocabulary, we compute it only over "A" and "B".

---

[12]To make interpretation easier, we balance the test labels by sampling one correct and one incorrect answer per question, resulting in a 50% random baseline (the training set is already balanced).

System Prompt:

> Your job is to evaluate if a proposed answer to an entity-centric question is correct.

User Prompt:

> Here is the question and the proposed answer.
>
> ```
>
> Question: {question}
>
> Proposed Answer: {answer}
> ```
>
> Is the proposed answer:
>
> A: CORRECT
>
> B: INCORRECT
>
> Just return the letters "A" or "B", with no text around it.

## A.9 Extended Definition of Knowledge

We now discuss the full definition of knowledge, which introduces the sanity-check expression $\gamma(\mathbf{q}; S_M)$ that handles *"implausible"* answer candidates (ones that are not in $\tilde{\mathbf{A}}(\mathbf{o})$). We omitted those details from the main definition (Definition 1) in order to make it easier to follow.

**Definition 3** (Extension to Knowledge of a Model w.r.t a Scoring Method)**.**
*As in Definition 1, we consider a model* $\mathbf{M}$*, and a fact* $\mathcal{F}$ *represented as a (subject, relation, object) triplet* $(\mathbf{s}, \mathbf{r}, \mathbf{o})$*, e.g., ("France", capital, "Paris"). We also denote the vocabulary of the tokenizer used by M with* $\mathcal{V}$*.*

*Then, in addition to* $\mathbf{Q}(\mathbf{s}, \mathbf{r})$*,* $\tilde{\mathbf{A}}(\mathbf{o})$ *and* $\mathbf{A}(\mathbf{o})$*, we define:*

- $\tilde{\mathbf{A}}_{\mathbf{M}}$*: The (infinite) set of all possible answers that* $\mathbf{M}$ *can produce, formally defined as* $\mathcal{V}^*$*. I.e., it is equal to the set of all finite sequences that can be formed using tokens from* $\mathcal{V}$*. It may include phrases such as "Paris", "Hello", "#%", etc.*

*We can then define the scoring function to be* $\mathbf{S}_M : \mathbf{Q}(\mathbf{s}, \mathbf{r}) \times \tilde{\mathbf{A}}_{\mathbf{M}} \to \mathbb{R}$ *instead* $\mathbf{S}_M : \mathbf{Q}(\mathbf{s}, \mathbf{r}) \times \tilde{\mathbf{A}}(\mathbf{o}) \to \mathbb{R}$*.*

*Next, we define the sanity-check indicator* $\gamma(\mathbf{q}; S_M)$ *that ensures that any plausible answer is scored above any non-plausible one:*

$$\gamma(\mathbf{q}; S_M) = \mathbb{I}\left(\forall \mathbf{a} \in \tilde{\mathbf{A}}(\mathbf{o}), \ \hat{\mathbf{a}} \in \tilde{\mathbf{A}}_{\mathbf{M}} \backslash \tilde{\mathbf{A}}(\mathbf{o}) \quad S_M(\mathbf{q}, \mathbf{a}) > S_M(\mathbf{q}, \hat{\mathbf{a}})\right)$$

*We then adjust the definition of the per-question score* $\mathbf{K}_{\mathbf{q}}(\mathbf{s}, \mathbf{r}, \mathbf{o}; \mathbf{S}_M)$ *to consider the sanity check:*

$$\mathbf{K}_{\mathbf{q}}(\mathbf{s}, \mathbf{r}, \mathbf{o}; \mathbf{S}_M) = \gamma(\mathbf{q}; S_M) \frac{1}{|\Omega(\mathbf{s}, \mathbf{r}, \mathbf{o})|} \sum_{(\mathbf{a}, \tilde{\mathbf{a}}) \in \Omega(\mathbf{s}, \mathbf{r}, \mathbf{o})} \mathbb{I}(\mathbf{S}_M(\mathbf{q}, \mathbf{a}) > \mathbf{S}_M(\mathbf{q}, \tilde{\mathbf{a}})) \tag{5}$$

We note that $\gamma(\mathbf{q}; S_M)$ should be consistently 1 for reasonable scoring methods, but we can also verify that by creating a focused challenge set.

## A.10 Hidden Knowledge in Larger Models

To make our findings reliable, we put special emphasis on sampling a large set of 1,000 answers per-question to approximate $\tilde{\mathbf{A}}(\mathbf{o})$ (see §3.2), which made our experiments computationally intensive. We use 7-9B models, which balance between model size and our compute budget. To provide initial evidence of hidden knowledge in larger models we run a smaller scale experiment with Qwen3-32B (Yang et al., 2025), using 200 sampled answers per-question. Figure 6 presents the results. For all relations, both the $\mathbf{K}$ and $\mathbf{K}^*$ measures are consistently higher when computed using the internal scoring function compared to any external one, suggesting that the hidden knowledge phenomenon persists even when scaling the number of parameters by a factor of ∼4. The average gap in $\mathbf{K}$ for Qwen is 12.5%, compared to 14%, 48%, and 57% for Llama, Mistral, and Gemma, respectively. Given that the gap for the larger Qwen model

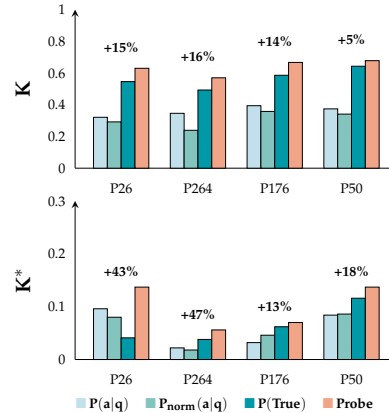

Figure 6: Avg. $\mathbf{K}$ and $\mathbf{K}^*$ for Qwen3-32B.

remains relatively close to that of the much smaller Llama, it remains an open question whether further scaling alone can be sufficient to reduce this gap, and which other factors affect the existence of it. Importantly, since 7–32B remains a commonly used capacity range, we argue that additional mitigation strategies beyond mere scaling up are needed to better expose model knowledge.

### A.11 $\mathbf{K_q}$ vs AUC-ROC

We note that $\mathbf{K_q}$ is conceptually related to the area under the ROC curve (AUC-ROC), as one interpretation of AUC-ROC is the probability that a randomly chosen positive instance is ranked higher than a randomly chosen negative one (Hanley & McNeil, 1982; Fawcett, 2006). One difference is that AUC-ROC averages over continuous thresholds, whereas our approach relies on direct pairwise comparisons, which also offers an intuitive interpretation of $\mathbf{K_q}$ as the fraction of correctly ranked answer pairs. More importantly, AUC-ROC is typically calculated across examples in a dataset, whereas $\mathbf{K_q}$ is computed separately *for each question*. This distinction is significant, as scoring methods' performance can vary substantially when comparing answers to the same question, as opposed to scoring question-answer pairs independently (Taubenfeld et al., 2025).

### A.12 Choosing the LLMs For Our Study

We chose the following three popular open-weight instruction-tuned LLMs for our study: Llama-3-8B-Instruct (Dubey et al., 2024),[13] Mistral-7B-Instruct (Jiang et al., 2023)[14] and Gemma-2-9B-Instruct (Team et al., 2024)[15]. We used the largest size that we could afford, as our experiments are compute heavy. We focus on instruct models as they are the ones that the users interact with, so the question of hidden knowledge is much more relevant to them. As discussed in §A.10, we also perform a smaller scale experiment with Qwen3-32B (Yang et al., 2025)[16] to provide evidence of hidden knowledge in larger capacity LLMs.

### A.13 Analysis of K Values When Manually Adding The Gold Answer to $\tilde{\mathbf{A}}(\mathbf{o})$

In Figure 7 we compare the $\mathbf{K}$ values between a setup that uses only the answers that were sampled as $\tilde{\mathbf{A}}(\mathbf{o})$ and our main setup (used in §4.1) where the gold is manually added to $\tilde{\mathbf{A}}(\mathbf{o})$. On average, in 64% of the cases we do not sample the gold answer, in which case adding the gold can change $\mathbf{K}$.

---

[13] https://huggingface.co/meta-llama/Meta-Llama-3-8B-Instruct

[14] https://huggingface.co/mistralai/Mistral-7B-Instruct-v0.3

[15] https://huggingface.co/google/gemma-2-9b-it

[16] https://huggingface.co/Qwen/Qwen3-32B

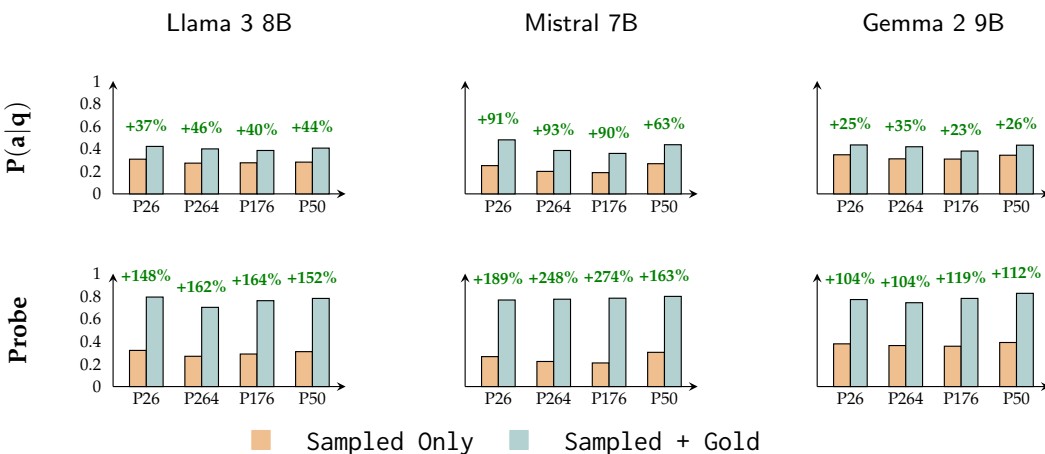

Figure 7: Comparison of average **K** values (see Equation (2)), under two conditions: *without* manually adding the gold (**Sampled Only**, left bars) and *with* manually adding the gold (**Force Gold**, right bars).

If we look at $\mathbf{P}(\mathbf{a}|\mathbf{q})$, we observe a consistent increase in **K** scores. We then further analyze the nature of the examples that lead to increase in **K**. In 97% of those cases, not only that the gold answer $a^{\mathrm{G}}$ was not sampled, but there was no other correct answer sampled, and thus **K** was manually set to 0. Accordingly, it is rather expected that in such cases adding the gold will lead it to a "win", increasing **K**. Yet, Figure 3 shows us that we never observe an increase in $\mathbf{K}^*$ for $\mathbf{P}(\mathbf{a}|\mathbf{q})$. In fact, in 7 out of 12 setups we even see a decrease in $\mathbf{K}^*$. In §4.2 we explain why $\mathbf{K}^*$ results are expected for $\mathbf{P}(\mathbf{a}|\mathbf{q})$ and we now show that some increase in **K** are also as one would expect.

When looking at **Probe**, not only it also shows a consistently increase in **K** scores, this increase is substantially higher than for $\mathbf{P}(\mathbf{a}|\mathbf{q})$. As discussed in §4.2, unlike $\mathbf{P}(\mathbf{a}|\mathbf{q})$, **Probe** shows consistent increase in $\mathbf{K}^*$.

### A.14    How K Affects Our Chances of Success in Inference Scaling?

We begin with a theoretical discussion on the relationship between our definition of knowledge and inference scaling. One interpretation of $\mathbf{K}(\cdot; S_M)$ is the probability of ranking a randomly chosen correct-vs.-incorrect answers pair correctly. Let $p = \Pr_{a \sim M}[a \in A(o)]$ be the probability to sample a correct answer from $M$. Then, when drawing $n$ i.i.d. samples, the probability that the top-ranked candidate is correct is:

$$\Pr[\text{Success}((s,r,o); n, p, S_M)] = \sum_{i=0}^{n} \underbrace{\binom{n}{i} p^i (1-p)^{n-i}}_{\substack{\text{Probability to sample } i \\ \text{correct and } (n-i) \text{ wrong}}} \cdot \underbrace{\left[ 1 - \overbrace{\left( 1 - \overbrace{K(s,r,o; S_M)^{n-i}}^{\substack{\text{Probability that one} \\ \text{correct answer out-ranks} \\ \text{all } (n-i) \text{ wrong ones}}} \right)^{i}}^{\substack{\text{Probability that all } i \text{ correct} \\ \text{answers fail to out-rank all wrong ones}}} \right]}_{\substack{\text{Probability that at least one correct} \\ \text{answer out-ranks all } (n-i) \text{ wrong ones}}}$$

$$(6)$$

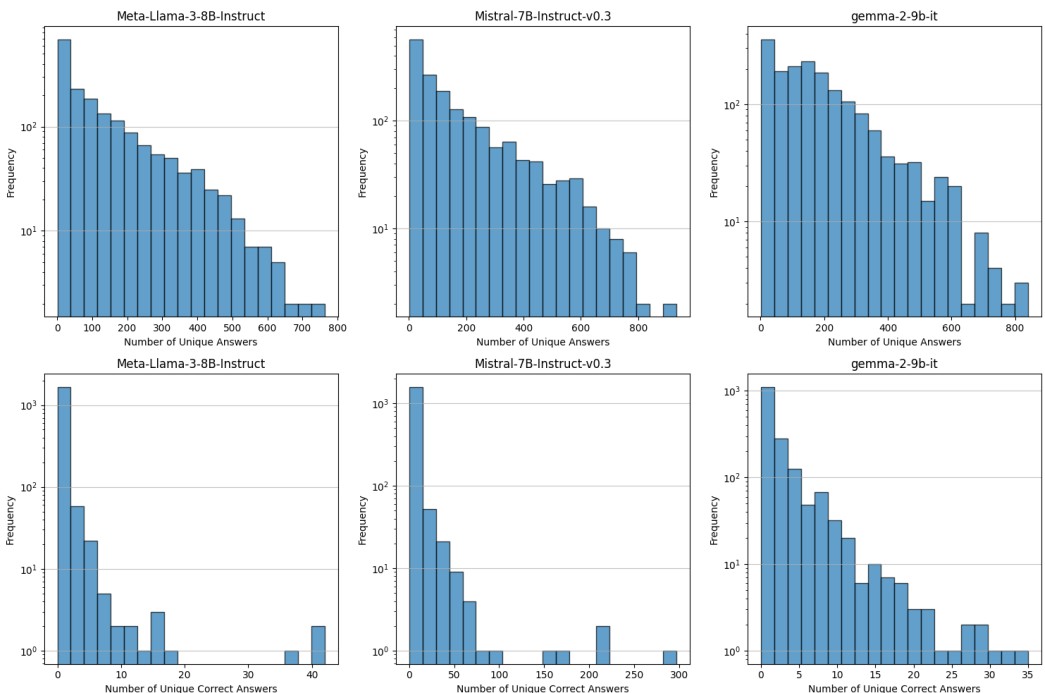

Figure 8: Statistics for the number of unique answers (top) and number of unique correct answers (bottom) per-questions. For a very small number of questions we observe a significantly high number of correct answers. This reflects cases where the model failed to provide a short-form answer and added additional diverse suffixes, e.g., *"Helen. This answer relates to the Greek mythology context."*. Those cases are rare and we can either filter-out such cases with a length threshold, or leave them and require the scoring functions to score them higher than wrong ones. Both options are legitimate and we validated that they do not affect our findings. For simplicity we report results without filtering.

It is evident that as $\mathbf{K}$ increases, so does $\Pr[\text{Success}(K)]$. Demonstrating a practical usefulness of our definition of knowledge: the more knowledge $M$ has according to our definition, the higher is its chance to benefit from inference scaling.

That said, it is also important to note that two models may exhibit identical inference scaling behavior despite significant differences in $K$. While, in principle, higher $\mathbf{K}$ and $\mathbf{K}^*$ scores increase the chances of success under inference scaling, they do not guarantee it. In fact, there are cases where a model $M1$ has a higher knowledge gap than another model $M2$, yet $M2$ benefits more from inference scaling. The key intuition is that $\mathbf{K}$ scores reward robustness across multiple correct phrasings, whereas inference scaling only requires the model to rank one correct answer above all incorrect ones.

To illustrate this, consider the following hypothetical example. Both $M1$ and $M2$ have the same five candidate answers: two correct $(c1, c2)$ and three incorrect $(w1, w2, w3)$. Suppose $M1$ ranks them as $(c1, w1, w2, w3, c2)$, and $M2$ as $(w1, c1, c2, w2, w3)$, where the leftmost is the top ranked answer. Under inference scaling, where the top-ranked answer is selected, $M1$ returns a correct answer $(c1)$ while $M2$ returns a wrong one $(w1)$. However, $M2$ achieves a higher $\mathbf{K}$ score. For $M1$ $\mathbf{K} = 3/10$ due to $c1 > w2$, $c1 > w2$ and $c1 > w3$, while for $M2$ $\mathbf{K} = 4/10$ due to $c1 > w2$, $c1 > w3$, $c2 > w2$ and $c2 > w3$. This illustrates how differences in K scores do not always predict differences in inference scaling outcomes. Same holds for $\mathbf{K}^*$. Suppose we now have a model $M3$ that ranks the answers as $(c1, c2, w1, w2, w3)$. Both $M1$ and $M3$ will succeed under inference scaling, but only $M3$ achieves $\mathbf{K}^* = 1$ (as all correct answers are ranked above all incorrect ones), while $M1$ gets $\mathbf{K}^* = 0$ (as $c2$ is ranked below several incorrect ones). This illustrates that even $\mathbf{K}^*$ may not fully explain inference scaling performance.

We believe that our choice to include rankings of different phrasings of the correct answer in our metric is important, even if it is not required for inference scaling. For example, a model might be used to verify responses, either in response to a user request or to generate a reward score, in which case recognizing alternative phrasings is crucial.

### A.15 Alternatives to QA Format

As we discuss in §2.1, we choose to work with a QA format but other alternatives exist. Specifically, we could examine the scores that $M$ assigns to claims reflecting the relevant fact. For instance, if our fact is *("Empire State Building", location, "NYC")*, then, instead of scoring alternative answers to a related question, such as *"Where is the Empire State Building located?"*, we could score different claims, measuring whether correct claims (e.g., *"The Empire State Building is located in NYC"*) score higher than contradicting claims (e.g., *"The Empire State Building is located in Paris"*). However, to ensure a meaningful comparison, claims must follow a fixed template and differ only in factual content. The QA format naturally facilitates this by providing an environment where this factor is easily controlled since we can keep the question fixed and compare alternative answers.

### A.16 Discussion On Possible Reasons And Mitigations For Hidden Knowledge

An interesting direction for future work is understanding the reasons for hidden knowledge. One hypothesis is that post-training sharpens the probability distribution, which may concentrate the probability mass on a specific answer. This may lead to a condition where correct answers that were plausible before post-training are assigned with low likelihood by the model even though the model encodes knowledge on their correctness. Another possible reason is that post-training may emphasize style, leading the model to choose fluent but less factual answers. A possible mitigation could be to encourage robustness to different phrasings by exposing the model to multiple correct answers during training, which would require modifying both the dataset labels and the optimization objective to jointly learn from them. Another solution may lie in adapting the model during test time using decoding methods that use internal signals to help surface known facts, e.g., Rimsky et al. (2024). Lastly, another possible direction is to address the issue during the reinforcement learning phase by designing reward signals that prioritize factuality over style. This is a challenging goal, as we do not want to compromise the stylistic fluency that makes these models so effective.

## B Limitations

A key limitation of our framework is its high computational cost. For each question and model, we must generate many candidate answers, label each candidate using an LLM judge, and then score them using multiple methods. This is also the reason we focused on 7–9B models and did not experiment with models of larger capacities. Another limitation (discussed in §5) is that our definition of knowledge does not consider knowledge of related facts. For example, knowing that Paris is the capital of France may also require knowing that Paris is located in France. We choose to leave this aspect out of scope and hope future work will explore corresponding extensions to our definition. Finally, a limitation of the $\mathbf{K}^*$ metric, which reflects *full* knowledge, is its sensitivity to labeling errors: an incorrect label assigned to a candidate answer can flip its score from 0 to 1 or vice versa. To address this issue, we put significant effort into ensuring high labeling quality by using an LLM judge, carefully designing its prompt, and performing extensive human evaluations to confirm its accuracy (see §A.3). This approach shows clear improvements over the commonly adopted exact-match method. Additionally, we introduce the *continuous* metric $\mathbf{K}$, which is less sensitive to labeling errors, as our main evaluation measure.

## C   Acknowledgements

This research is a collaboration between the Technion and Google Research. It was supported in part by a grant from Google. Part of this research was also supported by Open Philanthropy and an Azrieli Foundation Early Career Faculty Fellowship and part of it was funded by the European Union (ERC, Control-LM,101165402). Views and opinions expressed are, however, those of the author(s) only and do not necessarily reflect those of the European Union or the European Research Council Executive Agency. Neither the European Union nor the granting authority can be held responsible for them. Hadas Orgad was supported by the Apple AIML PhD fellowship.

