# OpenReview forum: "Inside-Out: Hidden Factual Knowledge in LLMs"
_colmweb.org/COLM/2025/Conference — COLM 2025_

### Official Review · Reviewer_LjuE · 2025-04-25

**Rating:** 7
**Confidence:** 4
**Ethics Flag:** 1

**Summary:**

The authors investigate the phenomenon of hidden knowledge within LLMs. Namely, they introduce a formal definition of "hidden knowledge" and provide a formal framework based on two distinct types of scoring functions: "external scoring" (using observable token-level probabilities) and "internal scoring'" (based on intermediate computations within the model). The paper applies this framework to evaluate three LLMs: LLaMA-3-8B, Mistral-7B, and Gemma-2-9B, demonstrating that using probes from their intermediate layers to score the answers consistently outperforms more traditional "external scoring" functions. The authors highlight significant gaps between the models' potential and their generative performance and perform a case study. Finally, they utilize these probes to get better results in Question Answering task for the same models.

**Questions To Authors:**

- The design of the experiment with probes (for internal scoring) is not fully explained. A major point of ambiguity is the selection of the development set for identifying the optimal layer. At line 222 you write: _"We train probes for all layers and choose the best layer based on a development set"_. But what is development set exactly?
- You choose local 14B model as a Judge. Did you consider using a better model via API, such as GPT-4-O?
- What does "oracle" refer to in Table 1?
- Are there additional case study examples beyond the one presented in Figure 4? It would be helpful to see more.
- **What is the accuracy of scoring on probes from other layers** of LLaMA, Mistral, and Gemma, aside from the best layer?

**Reasons To Accept:**

- The authors address an important problem of finding hidden knowledge inside LLMs and extracting it. This may be potentially helpful both to understand these models better and to improve their performance in the future.
- The formalism of the framework that the authors introduce looks reasonable.
- The design of data collection and sampling "plausible answers" also seems reasonable and well thought-out.
- The authors provide an insightful analogy between hidden knowledge in LLMs and a psycholinguistic phenomenon in humans, where people may understand words yet fail to retrieve them.

**Reasons To Reject:**

- **The paper lacks clarity in its presentation**. In particular, certain crucial details are scattered across several little appendices (A.5, A.6, A.7, A.10), which hinders the clarity of the paper. The most importantly, I suggest to integrate the general description of the probe training process (Appendix A.5) into the main text, and explain it better (refer to the specific questions for the authors below).
- The LLMs used in the paper are somewhat outdated. **The paper would benefit from adding at least one more recent model into the experiments**.
- The topic of hidden knowledge has already been explored in previous works from different perspectives. In particular, in the paper Listening to the Wise Few: Select-and-Copy Attention Heads for Multiple-Choice QA by Tulchinskii et al., it was found that answering MCQA tasks based on query-key interactions from specific attention heads of the LLaMA model outperforms the model itself on many MCQA tasks. In that work, they concluded that some parts of the model 'know' the answer, but this information sometimes is lost before it reaches the output layer. Additionally, there is a body of earlier work on extracting topological features from specific heads and layers of transformer models (Topological Data Analysis for Speech Processing by Tulchinskii et al., Acceptability Judgements via Examining the Topology of Attention Maps by Cherniavskii et al., Artificial Text Detection via Examining the Topology of Attention Maps by Kushnareva et al). In these works, topological features extracted from specific heads and layers also outperform the model itself in a number of task, serving as a sort of whimsical internal probe as well. **Summing up, the authors should better clarify the novelty of their work in light of other studies**. For example, you could emphasize a formal definition of hidden knowledge and a new method for identifying it as a clearer novelty of this particular work.

**UPD**: The authors have committed to improving the presentation, satisfactorily addressed my questions, and provided additional examples illustrating their point in the rebuttal. As a result, I have increased my score.

---

> ### Author Response · Authors · 2025-06-02
> **Authors Response: Part 1**
>
> Thank you for your thoughtful and encouraging feedback. We’re glad you found the problem we address important for both understanding and improving LLMs, that you considered our framework and data collection design well thought-out, and that you found the psycholinguistic analogy insightful. We address your concerns below and in a followup comment (due to response length limitations).
>
> ---
>
> ## Presentation Clarity of the Technical Details
>
> Thank you for this feedback. We tried to balance between providing the full technical details and not interrupting the reading flow too much, this was our motivation for extracting-out some technical details to appendices. After reading your feedback and reflecting, we agree that this extraction was too heavy and we commit to revise those parts in the camera-ready version.
>
> Concretely, we agree that the details in appendices A.5, A.6, A.7, A.10 should be discussed in the main text. The camera-ready version will contain: (1) a dedicated subsection or paragraph in section 3 that discusses the design of the probe (A.5), (2) a discussion on the alternatives to QA format (A.6) as part of the opening part of subsection 2.1 (before the definition of K), (3) elaboration on the statistic significance test (A.7) as part of the paragraph that discusses the “delta” threshold in subsection 3.2, and (4) a discussion on closely related metrics (A.10) at the end of subsection 2.1 (after the definition of K).
>
> The **development** and test sets are created jointly, and at the final stage we basically reserve 10% of the resulting data for development (lines 634-635). The data sizes are presented in Table 3, which we reference in line 170. We will provide more details on the data splits in the main paper moving them from A.1 (lines 625-649) to subsection 3.1. When discussing the probe details we will make sure to refer to this text.
>
> Thank you, we truly believe that these changes will improve the clarity of our paper.
>
>
> ---
>
> ## Adding One More Recent Model
>
> Following yours and reviewer’s 959U feedback, we will add results using **Qwen3-32B** to the camera-ready version. It is a new model with larger size, released just a month ago, on April 29, 2025 [1]. We cannot provide the full results during the rebuttal since our experiments are computationally intensive [2], and our limited access to compute resources makes it difficult to run experiments with new models, especially of a larger capacity. To provide clarity on the presence of hidden knowledge in newer frontier models during the rebuttal, we followed the suggestion by 959U and ran an initial experiment at a smaller scale [3]. The results indicate that Qwen3-32B also exhibits hidden knowledge, with about 34% gap between external and internal knowledge [4], as reflected in the table below. As in Figure 1, we report both the K and K* scores for each method, and compute the percentage difference between the best-performing external scoring function and our internal scoring function (Probe), which are all statistically significant.
>
> We are working on extending this experiment to the full-scale experimental setup to the camera ready version. We also wish to highlight that a big part of our contribution, beyond the empirical study, is laying the foundation for further studying this phenomenon by carefully designing our formal framework.
>
> | Scoring Method           | K    | K*   |
> |------------------|------|------|
> | `P(a\|q)`      | 0.41 | 0.08  |
> | `P_norm(a\|q)` | 0.40 | 0.09  |
> | `P(True)`            | 0.50 | 0.09  |
> | `Probe`         | 0.67 (**+34%**) | 0.13 (**+44%**) |
>
> [1] https://huggingface.co/Qwen/Qwen3-32B
>
> [2] We placed emphasis on producing reliable findings by sampling a large number of answers per-question to better approximate the answer space. This process involves generating about 500,000 unique answers per-model. Each of them needs to be labeled by our judge and scored using multiple methods.
>
> [3] We focused on a single, randomly selected relation (P176), and used smaller training and test sets of 800 and 200 questions, respectively. We sampled 200 answers (+1 greedy) per-question instead of 1,000.
>
> [4] Due to the smaller training set, the probe may be under-trained and thus reflect a lower bound on the knowledge gap.
>
> ---
>
> ## Clarify the Novelty
> Thank you for this feedback and suggestion, and for providing the additional references. We like the idea of emphasizing our formal definition and framework for studying hidden knowledge and we also share the view that this is where the novelty of our work lies compared to other studies. This is also why we chose to refer to this as “our core contribution” in lines 32-33. In the camera ready version of the paper, we will cite the additional references you provided and make sure to better highlight the novelty of our work as you suggested.

---

> > ### Author Response · Authors · 2025-06-02
> > **Authors Response: Part 2**
> >
> > ## Did you consider using a better model as a Judge via API?
> > Yes, we considered it, and we agree that using an API model such as GPT-4-O for this task is a good practice. However, due to the large-scale nature of our study we could not afford it. We needed to grade about 500,000 answers per-model (i.e., around 1.5M answers total). Instead, we chose to use a smaller model but took steps to ensure its quality. We carefully designed a program guided COT prompt that prompts the model to go through specific reasoning steps, which significantly improved the performance in early experiments. Next, we verified the final quality in a dedicated human evaluation. The details of this process are in A.3 (referenced from line 196). We will add an explicit clarification about why we did not use an API model in the camera-ready, and will also recommend using one if feasible.
> >
> > ---
> >
> > ## What does "oracle" refer to in Table 1?
> > In the Oracle scoring baseline, we assign each correct answer a score of 1 and each wrong answer a score of 0. Under inference scaling, where the top scoring answer is selected, it means that if a correct answer was sampled it will be also selected by the Oracle. The goal of this baseline is to provide an upper bound for a “perfect scoring method”. Thank you for pointing this out, we agree that the current description (in the caption of Table 1) is not clear enough and we will revise section 4.4 to more accurately describe this baseline.
> >
> > ---
> >
> > ## Probe Performance on Other Layers
> > The per-layer scores are available in this link: https://tinyurl.com/4czfvy2x. We observe that scores tend to improve and then mostly stabilize in the final two-thirds of the network, typically starting around layers 11–12 out of 32. This suggests that the upper layers encode higher-quality and more consistent representations compared to earlier layers. We will add the per-layer performance to the camera-ready version.
> >
> > ---
> >
> > ## Additional case study examples
> >
> > We will be happy to add more cases to the camera-ready version. We are also adding some here for you to review.
> >
> > In case study #1 (https://tinyurl.com/z3h29hec), the question is “Who is the author of Zweites Buch?” and the gold answer is “Adolf Hitler”. It was sampled by the model but it gets lower P(a|q) compared to “Hans Morgenthau”. Both “Adolf Hitler” and “Hans Morgenthau” are German-speaking historical figures associated with writings on foreign policy, which may explain the confusion. The Probe clearly ranks the correct answer the highest. This illustrates a case where the model may be confused by another plausible answer during generation yet encodes knowledge about the correct answer internally. Interestingly, even though P(True) usually outperforms P(a|q), in this case it performs poorly ranking many incorrect candidates above the correct one.
> >
> > We also include case study #2 (https://tinyurl.com/4z83tz3j), which illustrates a different scenario with a large number of unique sampled answers (262). Unlike the previous example, where the model leaned toward an incorrect but plausible answer, this case highlights a situation of high uncertainty: the model’s generation probabilities P(a∣q) are distributed across many candidates, and the verification score P(True) assigns similarly high scores to most of them. Despite this external uncertainty, the probe assigns the highest score to the correct answer.
> >
> > Case study #3 (https://tinyurl.com/yc347ryc) illustrates how the probe can successfully rank many possible phrasings of the correct answer above all incorrect ones.
> >
> > Lastly, since you expressed interest in newer models, we also add case study #4 (https://tinyurl.com/4jwnk3dp) and case study #5 (https://tinyurl.com/2u8mff8n) with examples from Qwen3-32B. We find case study #5 interesting. It describes the question “Which company is the German submarine U-60 (1939) produced by?” with the gold answer “Deutsche Werke”. It is evident that the model is confused by alternative answers “Howaldtswerke-Deutsche Werft” and “Deutsche Werft AG“. After further inspection we found that those are actually very closely related [5]. “Deutsche Werke” is a shipyard that was originally named “Deutsche Werft” and was renamed in 1918. Then, in 1968 it became a part of “Howaldtswerke-Deutsche Werft”. This illustrates that the model must be aware of the different namings of this shipyard but “externally” it struggles to recall which particular name was used during the production of the submarine. However, it is evident that  it encodes this knowledge internally as the probe ranks the correct answer higher than the rest by a large margin.
> >
> > [5] https://ww2db.com/facility/Deutsche_Werft_Hamburg/
> >
> >
> > ---
> >
> > ## Concluding Note
> >
> > Thank you again for the detailed feedback. We hope our response has addressed your concerns, and that you will reconsider your evaluation based on our response. If any points remain unclear, we would be glad to clarify and continue the discussion.

---

> > > ### Comment · Reviewer_LjuE · 2025-06-04
> > >
> > > Thank you for the insightful examples, explanations, and new experiments. Are there any cases where the model gives the correct answer to a question, but the internal probe produces an incorrect one? Or does the probe always provide the correct answer whenever the model does?

---

> > > > ### Author Response · Authors · 2025-06-06
> > > > **Response to followup question**
> > > >
> > > > Yes, there are also cases of such “probe losses”. For instance, if we look at cases where `P(a|q)` has a correct answer as the top ranked one, then in 75% of the cases the probe also ranks a correct answer as the top one and in 25% it is not. We can think of two explanations for why this occurs.
> > > >
> > > > ## Undertrained or Suboptimal Probe
> > > >
> > > > First, it is important to emphasize that our results should be interpreted as a lower bound on the model’s internal knowledge, since the probe may be suboptimal in our setting. We did not focus on optimizing the probe (see lines 206–208), so some probe errors may simply reflect limitations of the probe itself rather than a lack of internal knowledge in the model.
> > > >
> > > > To further examine this point, we compared the differences in K values for cases where the probe wins or loses against the model. Specifically, we define a probe loss as a case where `P(a|q)` ranks a correct answer highest but the probe does not, and a probe win as the opposite. The average difference in K values is 0.483 for probe wins, but only 0.046 for probe losses. This indicates that when the probe loses, it often does so by a very small margin, and often still ranks answers relatively well compared to `P(a|q)`, but just fails to select the top answer. See, for example: https://tinyurl.com/bdfsuxjm.
> > > >
> > > > ## Knowledge Encoded in Different Layers
> > > >
> > > > Second, in our setup we train the probe on a single layer (for simplicity), and we use the same fixed layer for all examples and relations [1] to maintain a single, well-defined internal function [2]. However, in practice, the model’s knowledge may be best encoded in different layers for different examples, or even distributed across multiple layers [3]. As a result, some probe losses may occur simply because our chosen layer is not optimal for a given case.
> > > >
> > > > To provide some initial evidence for this possibility, we examined probe performance across all layers. As mentioned above, the probe is correct in only 75% of the cases where `P(a∣q)` ranks the correct answer highest. However, when considering the best-performing probe across all layers, we found that in 92% of these cases, at least one probe correctly identifies the top answer. This suggests that the knowledge may still be encoded in the model but not optimally captured by the probe at the chosen layer.
> > > >
> > > > ---
> > > >
> > > > [1] Which we select based on performance on a development set.
> > > >
> > > > [2] An alternative we considered is predicting with a probe on each layer and reporting the maximum across layers. The disadvantage is that the internal function becomes ill-defined, as it is unclear which layer to use for a new example.
> > > >
> > > > [3] Which is another reason for why our results are lower bound.

---

> > > > > ### Comment · Reviewer_LjuE · 2025-06-09
> > > > >
> > > > > Thank you for the insightful examples and clear explanations. I have increased my score and look forward to seeing the improved version of your paper with all the details we discussed in the rebuttal incorporated in the camera-ready submission.

---

> > > > > > ### Comment · Reviewer_LjuE · 2025-06-09
> > > > > >
> > > > > > As a suggestion for future work, I would recommend investigating how different kinds of SFT (e.g., instruction tuning) influence the model’s ability to recover its inner knowledge.

---

### Official Review · Reviewer_T7HJ · 2025-05-08

**Rating:** 8
**Confidence:** 3
**Ethics Flag:** 1

**Summary:**

The paper explores and provides evidence for an interesting phenomenon in LLMs - hidden knowledge: the encoding of more factual knowledge in the model parameters than what expressed in their outputs. The author(s) build from explicit definitions such as that of knowledge (external and internal) and used them to define experiments aimed at testing for the presence of hidden knowledge. Using a QA task, they contrast the factual knowledge emerging from model outputs with that emerging from hidden states (as extracted using trained probes), demonstrating that through the latter the model can show awareness of the correct answer in spite of never generating it through large-scale sampling.

**Questions To Authors:**

"We train probes for all layers and choose the best layer based on a development set." -> What are your observations on what hidden layers tend to encode better factual knowledge and what encode more hidden knowledge? Were there any trends? There might be some interesting insights there to include in the paper.

Hidden knowledge is an interesting but puzzling phenomenon (since model outputs are based on hidden states, why is certain knowledge not leveraged for the final task?). I'm curious if you have any speculative hypotheses in terms of why this phenomenon would arise in LLMs and what could be done to align more external knowledge to the internal one that remains hidden.

**Reasons To Accept:**

- The paper is very well-written and well-structured. The research questions are well explained, with clear motivation for the experiments and the methodology adopted.
- The paper addresses a very interesting phenomenon (hidden knowledge), connected to an important topic for the NLP community (factual knowledge in LLMs). The results provide insights on how LLMs behave and can inspire new research on better leveraging the knowledge in LLMs.
- The paper provides convincing evidence for the existence of hidden knowledge in LLMs, as well as a formal framework to investigate it (in terms of definitions and formulas).

**Reasons To Reject:**

- The paper could benefit from a discussion - even if speculative in nature -  on what the results could entail for our understanding of LLMs, why this phenomenon of hidden knowledge would arise in the first place or how we could push LLMs to better use their knowledge. I encourage the author(s) to expand a bit on this in the camera-ready version of the paper.

---

> ### Author Response · Authors · 2025-06-02
> **Authors Response**
>
> Thank you for the thoughtful and positive feedback. We’re glad you found the paper well-written and well-structured, with clear motivation and methodology. We are especially happy that you found both the evidence for hidden knowledge in LLMs and the formal framework to investigate it convincing. We appreciate your recognition of the relevance of this phenomenon for the NLP community and hope the framework will support future research on factual knowledge in LLMs. We address your concerns below.
>
> ---
>
> ## Discussion on possible reasons for hidden knowledge
>
> Thank you for this feedback, we think it is a great idea, and we will add a discussion section with possible reasons for hidden knowledge and possible approaches to mitigate it to the camera-ready version of the paper.
>
> One possible hypothesis is that post-training techniques like supervised fine-tuning and RLHF often concentrate probability on one preferred token. As a result, answers that were plausible before post-training may receive very low probability, even though the model still encodes knowledge on their correctness. Another possible reason is preference learning with rewards that emphasize style or popularity, the model may choose fluent but less factual answers.
>
> These hypotheses have yet to be tested in future work. If they hold true, a possible mitigation could involve adapting the supervised fine-tuning phase to expose the model to multiple correct answer variants, thereby encouraging robustness to different phrasings. This would require modifying both the dataset labels and the optimization objective to jointly learn from them. Another possible direction is to address the issue during the reinforcement learning phase by designing better reward signals that prioritize factual correctness over stylistic appeal. This is a challenging goal, as we do not want to compromise the stylistic fluency that makes these models so effective. Lastly, another solution may lie in adapting the model during test time by developing new decoding paradigms that take into account internal signals to help surface known facts, even when they have extremely low `P(a|q)`. An example of a possible direction is [1].
>
> [1] Steering Llama 2 via Contrastive Activation Addition
>
>
> ---
>
> ## The per-layer performance
> We agree that deeper studying how and why the knowledge is encoded in different layers may definitely provide interesting insights. The per-layer scores are available in this link: https://tinyurl.com/4czfvy2x. We observe that scores tend to improve and then mostly stabilize in the final two-thirds of the network, typically starting around layers 11–12 out of 32. This suggests that the upper layers encode higher-quality and more consistent representations compared to earlier layers. Interestingly, we also see that there is often a tiny drop in the last layers, which are used during decoding. However, this drop is relatively small. Apart from that, we did not observe any interesting insights from analyzing the different layers. However, it was not our focus during the research, so it is possible that we missed some interesting insights there. We will add the per-layer performance to the camera-ready version and we will discuss further analyzing it as one of the possible directions for future work.
>
> ---
>
> ## Concluding Note
>
> Thank you again. We hope that we have addressed your concerns and would be happy to continue the discussion if you have any further questions.

---

> > ### Comment · Reviewer_T7HJ · 2025-06-10
> >
> > Thank you for your reply.
> >
> > I maintain my positive view on the paper and leave my score unchanged. I encourage the authors to incorporate in the paper the clarifications and discussion provided in the response.

---

### Official Review · Reviewer_6s8b · 2025-05-11

**Rating:** 5
**Confidence:** 4
**Ethics Flag:** 1

**Summary:**

This paper studies an important question on if LLM encodes more knowledge (internal knowledge) than it outputs. They propose a definition of internal knowledge and external knowledge. Concretely, to scoring internal knowledge, they train a linear classifier to predict correctness from the model’s hidden representations of question-answer pairs. For external knowledge, it’s based on the model’s token-level probabilities. Experimental results show that LLMs consistently encode more factual knowledge internally than externally (around 40%) based on the QA dataset.

Though, the paper studies an very important question, the original research question has been significantly simplified during the research formation, and have too much assumption and hypothesis (such as internal and external knowledge definition, and using QA to validate this), which still raise questions on if the findings from this simplified setting can used for the original question.

**Questions To Authors:**

* What is the implication for practical real-world applications?
* Will we have the same findings when using larger size LLMs?

**Reasons To Accept:**

* The paper studies an important research question, and provides definitions to some concepts to quantify the hidden knowledge in LLMs.
* The paper conducts experiments and demonstrates important findings on gap between internal knowledge and external knowledge.

**Reasons To Reject:**

* The original research question is significantly simplified, which raises questions on if the research findings are still applicable to the original question. For example, when training a scoring function, then a new knowledge will also be learnt during the training process, does the capability of the scoring function still represent the capability or knowledge of the original model?
* The paper only conducts experiments on QA task, the generalizability of the finding to other problems is unknown.

---

> ### Author Response · Authors · 2025-05-29
> **Part 1: Addressing Main Concerns**
>
> Thank you for your thoughtful feedback. We’re glad you find the research question important, appreciated the conceptual definitions we introduced to study hidden knowledge, and recognized the findings on the gap between internal and external knowledge as important.
>
> Your main feedback revolved around the simplification of the research question, which we distilled into two main concerns. We address them in detail below and would appreciate it if you could let us know if there's anything we may have missed.
>
> Due to response length limitations, and since we wanted to address your feedback thoroughly, we split our response into 2 parts. In this part we address your primary concerns and in the next part we answer your questions and summarise the response.
>
> ---
>
> ## Concern (1): The scoring function may rely on knowledge acquired during its training, potentially obscuring whether it truly reflects the original model’s capabilities.
>
> We took several steps to mitigate this knowledge contamination risk. Recall that each QA pair corresponds to a (`subject`, `relation`, `object`) triplet, where the question reflects a `subject` and `relation` while the answer reflects an `object` (see an example in lines 96-97). This allows us a high degree of control for factual knowledge that is learned by the probe vs the one that we are evaluating on. We first filter out test questions that appear in the training set (surprisingly this happened in our dataset). Second, we also filtered potential `subject`-`object` leaks for relations where the `subject` and the `object` are of the same type [1], by ensuring that train `subjects` do not appear as test `objects` and vice versa. We also considered filtering out overlapping `subjects` and `objects` across relations but our analysis showed that such cases were rare and did not pose a meaningful risk [2]. Lastly, we follow previous work and use a lightweight linear classifier as our probe [3,4,5,6,7,8], which limits the capacity of the internal scoring function to encode or introduce additional knowledge.
>
> In the camera-ready version, we will move this discussion to the main text (currently in lines 637-640 in Appendix A.1) and expand it to clearly articulate the potential risks, how our design mitigates them, and why we believe the scoring function remains a faithful probe of the original model’s knowledge.
>
> ---
>
> ## Concern (2): Findings are based on the QA task.
>
> We chose QA intentionally since our goal is to study the factual knowledge in LLMs and to ensure that we measure knowledge rather than other capabilities. To this end, we focus on knowledge that can be structured as (`subject`, `relation`, `object`) triplets. This choice is very common in existing work (see section 5, lines 316-329) and it helps control for confounding factors such as avoiding ambiguous or easily guessable answers and preventing overlaps between training and test examples (as discussed in Concern (1)). A common way to test the knowledge of factual triplets is to test the ability to predict the `object` given a `subject` and a `relation`, which can be naturally achieved via QA (see lines 96-97). There are alternative approaches, such as using claims, which we considered and discussed in Appendix A.6 (lines 768-784), but they do not have a clear advantage over our choice.
>
> We will revise this aspect in the camera-ready version to more clearly discuss both the advantages and limitations of this choice.
>
> ---
>
>
> [1] In our case it is `P26` (“married to”) where the subject and object are both “person”.
>
> [2] Specifically it could only happen for `P26` (“married to”) and `P50` (“author”) since those are the only two relations with the same `subject` type (“person”), but learning that a person is an author of something will not help to know who are they married to.
>
> [3] Discovering Latent Knowledge in Language Models Without Supervision
>
> [4] The geometry of truth: Emergent linear structure in large language model representations of true/false datasets
>
> [5] Cognitive Dissonance:Why Do Language Model Outputs Disagree with Internal Representations of Truthfulness?
>
> [6] The low dimensional linear geometry of contextualized word representations
>
> [7] Linear Adversarial Concept Erasure
>
> [8] LLMs Know More Than They Show: On the Intrinsic Representation of LLM Hallucinations

---

> > ### Author Response · Authors · 2025-05-29
> > **Part 2: Answers for Additional Questions and Response Summary**
> >
> > In this part we answer your questions and summarise the response.
> >
> > ----
> >
> > ## Question (1): What is the implication for practical real-world applications?
> > First, the ability to detect and measure hidden knowledge can guide the development of methods to reveal it, improving model accuracy and reliability (see lines 23–24). Second, It can help design safety measures, since hidden knowledge can be risky if it unexpectedly surfaces and reveals sensitive information (see lines 24–26). Lastly, although this is not the main focus of our work, we provide important insights into practical limitations on scaling test-time compute through repeated answer sampling in closed-book QA. Our findings offer strong evidence that significant performance gains remain inaccessible due to decoding limitations in current LLMs. We show that these limitations can prevent some correct answers from being sampled, even though, if sampled, they would be consistently ranked highest by the model. This provides strong motivation for future research: exploring alternative decoding mechanisms that allow for more effective sampling of answer candidates. Our results show evidence that successfully developing such sampling mechanisms will lead to increased ability to scale test time compute (see lines 73-76, 264-274 and 306-314).
> >
> > ----
> >
> > ## Question (2): Will we have the same findings when using larger size LLMs?
> >
> > This is a great question and a fascinating direction for future work. One possible hypothesis is that post-training may aggravate hidden knowledge since it may sharpen the tokens’ distribution in some ways, which may lead to a condition where correct answers that were likely are “pushed down”. This can also happen since post-training may encourage convergence to one specific answer which limits the model’s ability to “know” that there are alternative correct phrasings. This is just an hypothesis which is yet to be tested in future work. Due to the expensive computational nature of our experiments, and due to limited access to large-scale compute resources, we were unable to run experiments with larger models. Interestingly, we do find that even among models of comparable size, there are notable differences in their ability to fully expose their knowledge (see lines 231-233). This suggests that factors beyond size, such as architecture or training data, may also play a significant role.
> >
> > We are glad to see that our work sparks interest and believe that the formal framework that we proposed lays a good foundation for studying this phenomenon. We will add a speculative discussion on the possible reasons for hidden knowledge (as also suggested by reviewer T7HJ) and a call for future research to extend our study to a broader range of model sizes in order to better understand how scale affects a model’s capacity to surface internal knowledge.
> >
> > ----
> >
> > ## Summary
> >
> > Thank you for highlighting these important points. We hope that our response clarifies that we addressed them carefully in the design of our study but did not explain them clearly enough in the paper. We are committed to revising the paper accordingly.
> >
> > We hope that you will reconsider your evaluation based on our response and we are happy to continue the discussion in case you have further concerns.

---

> > > ### Author Response · Authors · 2025-06-02
> > > **Update Regarding Larger Models**
> > >
> > > We ran a *smaller scale* [1] experiment using **Qwen3-32B**: a larger and newer model, released just a month ago, on April 29, 2025 [2]. The results indicate that Qwen3-32B exhibits hidden knowledge, with about 34% gap between external and internal knowledge [3], as reflected in the table below. As in Figure 1, we report both the K and K* scores for each method, and compute the percentage difference between the best-performing external scoring function and our internal scoring function (Probe), which are all statistically significant. You can view a qualitative example, in the same format as our case study in Figure 4, in this link: https://tinyurl.com/4jwnk3dp.
> > >
> > > | Scoring Method           | K    | K*   |
> > > |------------------|------|------|
> > > | `P(a\|q)`      | 0.41 | 0.08  |
> > > | `P_norm(a\|q)` | 0.40 | 0.09  |
> > > | `P(True)`            | 0.50 | 0.09  |
> > > | `Probe`         | 0.67 (**+34%**) | 0.13 (**+44%**) |
> > >
> > >
> > > We are working on extending this experiment to the full-scale experimental setup to the camera ready version, and we will also add a call for future work to further explore the effects of model scale on its ability to expose its internal knowledge. We could not perform a full-scale experiment during the rebuttal since our experiments are computationally intensive [4], and our limited access to compute resources makes it difficult to run experiments with new models, especially of a larger capacity.
> > >
> > > We hope this experiment provides further clarity on the presence of hidden knowledge in larger frontier models.
> > >
> > > ---
> > >
> > > [1] We focused on a single, randomly selected relation (P176), and used smaller training and test sets of 800 and 200 questions, respectively. We sampled 200 answers (+1 greedy) per-question instead of 1,000.
> > >
> > > [2] https://huggingface.co/Qwen/Qwen3-32B
> > >
> > > [3] Due to the smaller training set, the probe may be under-trained and thus reflect a lower bound on the knowledge gap.
> > >
> > >
> > > [4] We placed emphasis on producing reliable findings by sampling a large number of answers per-question to better approximate the answer space. This requires generating about 500,000 unique answers per-model. Each of them needs to be labeled by the judge and scored using multiple methods.

---

> > > > ### Comment · Reviewer_6s8b · 2025-06-03
> > > > **Reply**
> > > >
> > > > Thank you for responding to my review. As part of my reviewer's responsibility, I want to point out that my primary concern remains unaddressed. While I understand the study demonstrates we can achieve better factual knowledge prediction using internal parameters compared to external token probabilities using lightweight training-based probing, it's not very convincing to the claim that
> > > >
> > > > > "LLMs encode more factual knowledge in their parameters than what they express in their outputs."
> > > >
> > > > The response acknowledges that the lightweight method limits the capacity to introduce additional information.
> > > >
> > > > > "Lastly, we follow previous work and use a lightweight linear classifier as our probe [3,4,5,6,7,8], which limits the capacity of the internal scoring function to encode or introduce additional knowledge."
> > > >
> > > > However, it would be necessary to quantify the amount of additional information introduced during the lightweight training process. I would be willing to raise my rating if the paper revised its claim to be more precise and better aligned with the demonstrated results, quantify the faithfulness of the probe method.

---

> > ### Author Response · Authors · 2025-06-05
> > **Followup on Preventing Test-Train Leakage**
> >
> > Thank you for engaging in a discussion and for willingness to raise your rating.
> >
> > You're absolutely right: the fact that the probe is lightweight or linear does not guarantee the absence of train-test leakage. The final sentence in our previous response was not directly relevant to this concern. A simple probe only limits the capacity to memorize, but it does not rule out the possibility of leakage.
> >
> > We will now clarify more precisely how we prevent leakage.
> >
> > The core idea is that we ensure that the factual information that is encoded in the training data is not helpful for answering test questions. We adopt this practice from previous work by Gekhman et al., 2024 [1]. We will now list qualitative and quantitative arguments for why it works in our setup. We will use the relation P26 as a running example.
> >
> > ## The qualitative argument
> >
> > The fact that we work with QA pairs that can be mapped to specific (`subject`, `relation`, `object`) triplets allows us to control for the knowledge encoded in the training vs test sets. To qualitatively illustrate that, we attach 50 examples of questions from P26 with examples of correct and wrong answers from our dataset in this link: https://tinyurl.com/245a4t6j. The question is always of the form **“Who is** `{person_name}` **married to?”** and the answer is always another `person_name`.
> >
> > We verify that the training and test sets have disjoint person names and this results in a condition where **learning training examples is not useful for classifying test questions**. E.g., knowing that “Fiorella Kostoris” is married to “Tommaso Padoa-Schioppa” and not “Umberto Bossi” will not be helpful for classifying whether “Louis-Do de Lencquesaing” is married to “Caroline Champetier” or “Carole Fuentes”.
> >
> >
> > ## How can we quantify this?
> >
> > To empirically demonstrate that the probe's performance is not due to memorizing training examples, we compared models trained on different representations of the same input text. Let `f(q, a) = f"{q} {a}"` denote the concatenation of a question `q` and an answer `a`. In the probe setup, we train a classifier over a hidden representation `H(f(q, a))` extracted from the LLM. As additional baselines, we trained classifiers on two alternative representations:
> >
> > - TF-IDF features derived from the same input, denoted as `TFIDF(f(q, a))`.
> >
> > - Embedding-based features obtained by mean-pooling the token embeddings from the LLM's input embedding layer, denoted as `EMBED_MEAN(f(q,a))`.
> >
> > For both feature types, we train a logistic regression classifier as well as two fully connected neural networks with different depths [2].
> >
> > As shown in the table below, all these classifiers perform near chance. This provides evidence that facts or knowledge present in the training questions and answers, as well as any superficial textual patterns, cannot be leveraged by these models to generalize to the test set. Therefore, the probe’s performance is likely driven by information encoded in the model’s internal representations, rather than by memorizing the text or explicit facts from the training data.
> >
> >
> > | Input Representation         | Model                         | Train Accuracy (%) | Test Accuracy (%) |
> > |-----------------------------|-------------------------------|--------------------|-------------------|
> > | `-`                         | Random                        | -                  | 50.0              |
> > | `TFIDF(f(q,a))`             | Logistic Regression           | 95.6               | 50.5              |
> > | `TFIDF(f(q,a))`             | MLP (256, 256)                | 99.9               | 50.7              |
> > | `TFIDF(f(q,a))`             | MLP (512, 512, 256, 128)      | 99.9               | 50.2              |
> > | `EMBED_MEAN(f(q,a))`        | Logistic Regression           | 98.8               | 49.8              |
> > | `EMBED_MEAN(f(q,a))`        | MLP (256, 256)                | 99.9               | 49.1              |
> > | `EMBED_MEAN(f(q,a))`        | MLP (512, 512, 256, 128)      | 99.9               | 50.9              |
> > | `h(f(q,a))`                 | Logistic Regression           | 99.9               | **64.0**          |
> >
> >
> >
> >
> > ## Concluding note
> >
> > We hope this addresses your concern and will revise the paper accordingly. If any aspects remain unclear, we’re happy to continue the discussion.
> >
> > ---
> >
> > [1] Does Fine-Tuning LLMs on New Knowledge Encourage Hallucinations?
> >
> > [2] We conducted a quick experiment using the P26 training set and test sets and Qwen3-32B as the LLM. To make interpretation easier, we balance the test labels by sampling one correct and one incorrect answer per question, resulting in a 50% random baseline (the training set is already balanced).

---

> > > ### Author Response · Authors · 2025-06-08
> > > **Quick Followup on Addressing Your Main Concern**
> > >
> > > Dear reviewer,
> > >
> > > Thank you again for your valuable comments.
> > >
> > > Since the discussion period is still open, we wanted to briefly check in to see whether our latest response has addressed your concern. If anything remains unclear, we’d be happy to provide further clarification.
> > >
> > > If our response has resolved your concern, we hope you might consider updating your evaluation accordingly.
> > >
> > > Best regards,
> > >
> > > Authors

---

> > > > ### Comment · Reviewer_6s8b · 2025-06-10
> > > >
> > > > Thanks for the response! Read the response, adjusted the score, I do not have other questions. Thanks!

---

### Official Review · Reviewer_959U · 2025-05-12

**Rating:** 7
**Confidence:** 3
**Ethics Flag:** 1

**Summary:**

The paper introduces a formal and measurable notion of hidden factual knowledge in large language models (LLMs) and shows that internal signals in the models’ hidden states rank correct answers better than common probability-based external signal, revealing a knowledge gap and cases where the model “knows” an answer. The experiments are conducted on three LLMs (Llama-3-8B-Intruct, Mistral-7B-Instruct, Gemma-2-9B-Instruct) using specifically constructed closed-book QA dataset.

**Questions To Authors:**

Questions based on reasons to reject:
- **(Q1)**: Did you attempt even a small-sample study on 30B or bigger models? If not, do you think the results will be qualitatively different?
- **(Q2)**: Do you think that your results will be applicable to other datasets for increasing test-time compute?
- **(Q3)**: Could you provide more results, showing the reliability of constructed linear classifier like in (R3)?

Other question:
- **(Q4)** Have you experimented with alternative decoding (e.g., nucleus sampling, temperature sweep, MCTS) to see if “never-generated” but known answers can be surfaced?
- **(Q5)** I don't completely understand, why the increase for Gemma is so small compared to other models in Table 1, while Gemma has the biggest gap between internal and external knowledge. Could you please comment on this?

**Reasons To Accept:**

- **(A1)** Convincing methodological setup: authors provide precise definition distinguishing internal vs external knowledge and metrics to quantify them. They compare multiple external scores, using 1 000 diverse samples, an LLM judge with >99 % estimated accuracy, and statistical significance thresholds.
- **(A2)** Strong empirical evidence: across all settings the probe outperforms every external scorer with significant margins. Hidden-knowledge gaps differ across models (e.g. 62% Gemma vs 14% Llama), but it is statistically significant.
- **(A3)** Clear presentation and structure: the materials are presented in a very clear and structured way, and all experiments are accompanied by clear illustrations.
- **(A4)** Practical takeaway: authors provides a way to rank answers based on probes and demonstrates a 12% average accuracy boost in closed-book QA. However, there is some arguable moments (see R2)

**Reasons To Reject:**

- **(R1)** Limited scale: only 7B-9B models are tested, so conclusions may not hold for frontier 70B-175B models. It would made results more convincing if at least one 70B model will be included.
- **(R2)** Limited practical availability: based on my understanding, authors provide results on increasing test-time compute only on their own dataset. I believe, it would be more interesting to see other (probably even out-of-domain) datasets and corresponding results.
- **(R3)** Insufficient analysis of internal knowledge probes: authors didn't analyse the constructed probe classifiers to inspect its reliability – i.e. metrics on training set, the visualisation of activations manifold and the analysis of metrics per layer.

---

> ### Author Response · Authors · 2025-06-02
> **Authors Response: Part 1**
>
> Thank you for your thoughtful feedback. We’re glad you found the methodological setup convincing, the empirical results strong, the presentation clear and well-structured, and the practical takeaway valuable. We address your concerns below and in a followup comment (due to response length limitations).
>
> ---
>
> ## (R1) / (Q1): Studying Larger Models
> We followed your suggestion and ran a *smaller scale* experiment using **Qwen3-32B**: a larger and newer model, released just a month ago, on April 29, 2025 [1]. We focused on a single, randomly selected relation (`P176`), and used smaller training and test sets of 800 and 200 questions, respectively.  We sampled 200 answers (+1 greedy) per-question instead of 1,000. The results indicate that Qwen3-32B exhibits hidden knowledge, with about 34% gap between external and internal knowledge [2], as reflected in the table below. As in Figure 1, we report both the K and K* scores for each method, and compute the percentage difference between the best-performing external scoring function and our internal scoring function (Probe), which are all statistically significant. You can view a qualitative example, in the same format as our case study in Figure 4, in this link: https://tinyurl.com/4jwnk3dp.
>
> | Scoring Method           | K    | K*   |
> |------------------|------|------|
> | `P(a\|q)`      | 0.41 | 0.08  |
> | `P_norm(a\|q)` | 0.40 | 0.09  |
> | `P(True)`            | 0.50 | 0.09  |
> | `Probe`         | 0.67 (**+34%**) | 0.13 (**+44%**) |
>
>
> We are working on extending this experiment to the full-scale experimental setup to the camera ready version, and we will also add a call for future work to further explore the effects of model scale on its ability to expose its internal knowledge. We could not perform a full-scale experiment during the rebuttal since our experiments are computationally intensive [3], and our limited access to compute resources makes it difficult to run experiments with new models, especially of a larger capacity.
>
> We hope this experiment provides further clarity on the presence of hidden knowledge in larger frontier models.  We also wish to highlight that a big part of our contribution, beyond the empirical study, is laying the foundation for further studying this phenomenon by carefully designing our formal framework.
>
> [1] https://huggingface.co/Qwen/Qwen3-32B
>
> [2] Due to the smaller training set, the probe may be under-trained and thus reflect a lower bound on the knowledge gap.
>
> [3] We placed emphasis on producing reliable findings by sampling a large number of answers per-question to better approximate the answer space. This requires generating about 500,000 unique answers per-model. Each of them needs to be labeled by the judge and scored using multiple methods.
>
> ---
>
> ## (R2) / (Q2): Regarding the Test-Time Compute Results
> We agree that it would be interesting to extend our inference-scaling experiments to other datasets or domains. Since the main focus of the paper is not on proposing a practical method for boosting performance, but rather to define and measure hidden knowledge, we chose to conduct only a preliminary experiment and leave such extensions out of scope. We also want to emphasize that, beyond the accuracy boost, one of the main purposes of this experiment is to illustrate the *negative* implications of our finding on the extreme case of hidden knowledge from section 4.2. Specifically, we wish to highlight the existence of a significant unused potential to scaling test-time compute that is caused by the limitations that we discovered in the ability to sample known answers, and to motivate future research on advanced decoding mechanisms. Thank you for this feedback, we will revise the camera-ready version to clarify both the scope and the main purpose of this experiment.
>
> ---
>
> ## (R3) / (Q3): Additional Analysis of the Probe
> We observe that scores tend to improve and then mostly stabilize in the final two-thirds of the network, typically starting around layers 11–12 out of 32. This suggests that the upper layers encode higher-quality representations. In some cases we see a relatively small drop in the last layers compared to middle ones. We will add the per-layer performance to the camera-ready version, you can view them in this link: https://tinyurl.com/4czfvy2x. The training accuracy is very high, almost always around 95-99%, which suggests that the data is linearly separable. We explored several regularization techniques, but none led to meaningful improvements on the development set.
>
> Our goal was to use the probe as a tool to demonstrate hidden knowledge and thus it was sufficient for it to outperform external scorers. We applied standard techniques from prior work (examples: https://tinyurl.com/3x8uj4fz) using a simple and lightweight linear classifier, which limits its capacity to encode or introduce additional knowledge or learn spurious features.
>
> We will include probe diagnostic metrics in the camera-ready version.

---

> > ### Author Response · Authors · 2025-06-02
> > **Authors Response: Part 2**
> >
> > ## (Q4) Have you experimented with alternative decoding (e.g., nucleus sampling, temperature sweep, MCTS) to see if “never-generated” but known answers can be surfaced?
> >
> > Yes, we tried all of the mentioned techniques except MCTS. The core problem is that while these methods can increase diversity, they all sample from the same token-level distribution. They rely on the model’s next-token probabilities, which determine `P(a∣q)`, the model’s probability of generating answer `a` given question `q`. Whether using temperature, nucleus sampling, or even search-based methods like MCTS, decoding is guided by this distribution. So if `P(a∣q)` is low, the chance of generating `a` remains low. We briefly discuss this in Footnote 6. This is also our motivation for directly quantifying cases where the model has perfect knowledge of a fact but an extremely low `P(a∣q)`, see lines 266-270.
> >
> > We think that the solution may lie in developing completely new decoding paradigms that take into account internal signals to help surface known facts, even when they have extremely low `P(a|q)`. An example of a possible direction is [4]. We will revise the paper to better clarify this point.
> >
> > [4] Steering Llama 2 via Contrastive Activation Addition (https://aclanthology.org/2024.acl-long.828.pdf)
> >
> > ---
> >
> > ## (Q5) I don't completely understand, why the increase for Gemma is so small compared to other models in Table 1, while Gemma has the biggest gap between internal and external knowledge. Could you please comment on this?
> >
> > This is a very important and interesting question! While, in principle, higher K and K* scores generally increase the chances of success under inference scaling, they do not guarantee it. In fact, there are cases where a model `M1` has a higher knowledge gap than another model `M2`, yet `M2` benefits more from inference scaling. The key intuition is that K scores reward **robustness across multiple correct phrasings**, whereas inference scaling only **requires the model to rank one correct answer** above all incorrect ones.
> >
> > To illustrate this, consider the following hypothetical example. Both `M1` and `M2` have the same five candidate answers: two correct (`c1`, `c2`) and three incorrect (`w1`, `w2`, `w3`). Suppose `M1` ranks them as (`c1`, `w1`, `w2`, `w3`, `c2`), and `M2` as (`w1`, `c1`, `c2`, `w2`, `w3`), where the leftmost is the top ranked answer. Under inference scaling, where the top-ranked answer is selected, `M1` returns a correct answer (`c1`) while `M2` returns a wrong one (`w1`).  However, `M2` achieves a higher K score. For `M1` K=**3**/10 due to `c1`>`w2`, `c1`>`w2` and `c1`>`w3`, while for `M2` K=**4**/10 due to `c1`>`w2`, `c1`>`w3`, `c2`>`w2` and `c2`>`w3`. This illustrates how differences in K scores do not always predict differences in inference scaling outcomes. Same holds for K*. Suppose we now have a model `M3` that ranks the answers as (`c1`, `c2`, `w1`, `w2`, `w3`). Both `M1` and `M3` will succeed under inference scaling, but only `M3` achieves K* = 1 (as all correct answers are ranked above all incorrect ones), while `M1` gets K* = 0 (as `c2` is ranked below several incorrect answers). This illustrates that even K* may not fully explain inference scaling performance.
> >
> > We do believe that our choice to capture different phrasings is important. Inference scaling is just one use case. Another use case example may be using a model to verify responses, or even generating a reward score, in which case knowing different answer phrasings is crucial. Verification is needed not only for model responses though, a user may also directly ask the model if some claim is correct.
> >
> > Thank you for raising this question, we will revise the paper to include a more detailed discussion on the relation between our robustness-focused measures of hidden knowledge and the success in inference scaling.
> >
> > ---
> >
> > ## Concluding Note
> >
> > Thank you again for the detailed feedback. We hope that we addressed your concerns, and that you will reconsider your evaluation based on our response. In case any aspects require further clarification, we will be happy to continue the discussion and provide the necessary details.

---

> > > ### Comment · Reviewer_959U · 2025-06-04
> > >
> > > Thank you for the detailed and thoughtful response, and especially for taking the extra effort to include results from a larger model and provide additional analysis of the probe.
> > >
> > > Regarding the probe diagnostics: in the per-layer results (e.g., for Llama3-8B), some relations such as P264 seem to yield relatively low probing accuracy across layers (often under 80%). Since your main metric (e.g., K in Figure 2) relies on the quality of internal probing, is it fair to assume that in such cases your reported K scores may actually underestimate the true extent of internal knowledge, simply due to the probe’s limited reliability? If I am mistaken, then could you clarify this?

---

> > > > ### Author Response · Authors · 2025-06-05
> > > > **Response to followup question**
> > > >
> > > > Thank you for raising this point. It is something we should clarify more explicitly in the paper.
> > > >
> > > > First of all, you are correct to observe that if the probe is undertrained or suboptimal, then we will underestimate the true extent of internal knowledge. The performance on `P264` is indeed lower than for other relations (though still significantly above a random baseline), which could indicate that the probe is undertrained on this relation. However, we want to emphasize that low probe performance on a particular relation does not necessarily mean that the probe is undertrained; it can also reflect genuinely limited internal knowledge for that specific relation. Since we cannot disentangle these two factors, the reported results should indeed be interpreted as a lower bound. It is possible that the probe is suboptimal in our setting, as we did not focus on optimizing it (see lines 206–208).
> > > >
> > > > Please let us know if this addresses your question. We will revise the paper to clarify this and explicitly note that the reported scores should be interpreted as lower bounds.

---

> > ### Author Response · Authors · 2025-06-08
> > **Follow-Up on Rebuttal Discussion**
> >
> > Dear reviewer,
> >
> > Thank you again for your detailed and valuable comments. We’ve aimed to address all your concerns in our rebuttal and believe that the resulting revisions will improve the quality of our work.
> >
> > Given that we still have time for discussion, we wonder if there are any aspects that you feel remain unaddressed. Please let us know, as we’ll be happy to provide further clarification as needed.
> >
> > If our responses have resolved your concerns, we hope you might consider updating your evaluation accordingly.
> >
> > Best,
> >
> > Authors

---

> > > ### Comment · Reviewer_959U · 2025-06-09
> > >
> > > Thank you for your comments and clarifications, and sorry for the late response. I am raising the score accordingly to 7, and I am looking forward for updates on your paper!

---

### Decision · Program_Chairs · 2025-07-08

**Decision:**

Accept

**Comment:**

This paper investigates whether an LLM encodes moore knowledge internally (in embedding) than its generated tokens. All reviewers find the problem interesting and important. The technique proposed in the paper to estimate internal and external knowledge (using trained linear classifier based on latent embeddings, and using model's token level probability) are reasonable and valid. The paper's experimental results show that LLM encoder more factual knowledge internally than externally. Most reviewers have a major concern regarding the evaluated model size. The author addressed the concern by adding QWen 32B result. I highly encourage authors to add those in the revision.

The pape could be improved if the authors add: 1) a discussion on tasks other than QA; 2) probing other layers in LLaMA, Mistral, Gemma etc; 3) reliability analysis.